



# The impact of nitrogen and sulphur emissions from shipping on exceedances of critical loads in the Baltic Sea region

Sara Jutterström[1], Filip Moldan[1], Jana Moldanová[1], Matthias Karl[2], Volker Matthias[2] and Maximilian Posch[3]

[1]IVL Swedish Environmental Institute, Box 53021, SE-400 14 Gothenburg, Sweden
[2]Chemical Transport Modelling, Helmholtz-Zentrum Geesthacht, D-21502 Geesthacht, Germany,
[3]International Institute for Applied System Analysis (IIASA), Schlossplatz 1, A-2361 Laxenburg, Austria

*Correspondence to*: Sara Jutterström (sara.jutterstrom@ivl.se)

**Abstract.** The emissions of nitrogen (N) and sulphur (S) species to the atmosphere from shipping significantly contribute to S and N deposition near the coast, and to acidification and/or eutrophication of soils and freshwaters. In the countries around the Baltic Sea the shipping volume and its relative importance as a source of emissions are expected to increase if an efficient regulation would not be implemented. To assess the extent of environmental damage due to ship emissions for the Baltic Sea area, the exceedance of critical loads (CLs) for N and S has been calculated for the years 2012 and 2040. The paper evaluates the effects of several future scenarios including the implementation of NECA and SECA (Nitrogen resp. Sulphur Emission Control Areas). The implementation of NECA and SECA caused a significant decrease in exceedance of critical loads for N as a nutrient while the impact on the – already much lower – exceedance of critical loads for acidification was less pronounced. The relative contribution from Baltic shipping to the total deposition decreased from 2012 to the 2040 scenarios for both S and N. In contrast to exceedances of CLs for acidification, shipping still has an impact on exceedances for eutrophication in 2040.

## 1 Introduction

Anthropogenic emissions of sulphur (S) and nitrogen (N) species to the atmosphere and subsequent deposition have led to severe environmental problems such as acidification of soils and lakes, impacting freshwater ecosystems and damaging forests (e.g., Grennfelt, 2018). N deposition can also enhance eutrophication on land and water, increasing the risk of ecosystem damage and changes in biodiversity (e.g., Bobbink et al., 2010).

The main source of anthropogenic S to the atmosphere is through combustion of S-containing fossil fuels, the S reacting during combustion with oxygen, forming sulphur oxides ($SO_x$), mainly sulphur dioxide ($SO_2$) which can be further oxidized in the atmosphere forming sulphuric acid. This oxidation takes place either in the gas phase through a reaction with an OH radical or through heterogeneous oxidation in cloud or fog droplets or aerosol particles. Gas-phase sulphuric acid contributes to the formation of particulate matter and is a key species in new particle formation. Most of the deposition of S is in the form of wet and dry deposition of particulate sulphate (PM-$SO_4$) and dry deposition of $SO_2$. Combustion is also a source of oxides of



nitrogen (NOx) to the atmosphere. NOx-species are to a large extent produced from the reaction of molecular nitrogen and oxygen during the combustion, especially at high temperatures and an excess of oxygen. While in a fresh combustion exhaust, oxides of nitrogen are dominated by NO (typically 85-95 %), in the atmosphere NO quickly reacts with ozone and the NOx mixture is dominated by $NO_2$. Atmospheric oxidation of NOx takes predominantly part in the gas phase through reaction of

$NO_2$ with an OH radical, forming nitric acid ($HNO_3$). Other oxidation channels involve nitrogen pentoxide ($N_2O_5$, night-time oxidation channel), peroxyacetyl nitrates (PAN), peroxynitric acid ($HNO_4$) or nitrous acid (HONO, heterogeneous oxidation channel). Oxidised N is then deposited in form of gaseous $HNO_3$ and to some extent $NO_2$, PAN, $N_2O_5$ and other species. The majority of oxidised N is deposited as particulate nitrate ($NO_3^-$) through wet and dry deposition processes. Another important N species is ammonia ($NH_3$). The largest source of $NH_3$ is agriculture. In the atmosphere ammonia reacts readily with both

$HNO_3$ and $H_2SO_4$ forming particulate ammonium sulphate and nitrate (($NH_4$)$_2SO_4$, $NH_4NO_3$). Part of ammonia is also deposited directly through gas deposition. As the marine air-masses carrying shipping emissions arrive at coastal areas with agricultural activities, these reactions cause increased particle formation and deposition of these species in these areas.

During the last decades emissions of S and N air pollutants from land-based sources have been substantially reduced over

Europe. S emissions in Europe peaked around 1980 and have decreased 91% between 1990 and 2016 in the EU; and N emissions have also decreased during this time period, although not quite as dramatically ($NO_x$ emissions dropped by 58% and $NH_3$ emissions by 23%, EEA Report, 2018). Due to the emission reductions of S and N there has been a significant improvement of the acidification status of European ecosystems since the 1990s (De Wit et al., 2015), but also eutrophication has decreased.


The Convention on Long-range Transboundary Air Pollution (LRTAP Convention, www.unece.org/env/lrtap) under the United Nations Economic Commission for Europe (UNECE) was signed in 1979 and entered into force in 1983 with the purpose to limit and find solutions to cross border air pollution problems within Europe. Under the LRTAP Convention the critical load (CL) concept was adopted and further developed. A CL is a deposition threshold for a given pollutant, above

which unacceptable damage may occur, in the long run. The development of methodologies focussed on S and N deposition to derive CLs for acidity, due to S and N, and eutrophication, due to N. Since 1994, Protocols to the LRTAP Convention to reduce air pollution due to S and N deposition have been "effects-based", i.e. the sensitivity of ecosystems, expressed by their CLs, has been guiding the emission reduction agreements, with the ultimate aim reducing depositions below the CLs.

Reductions of S emissions in Europe have led to substantial decreases in the exceedances of CL for acidity (CLaci) both in area and Average Accumulated Exceedance (AAE, Posch et al., 2001). The exceeded area for CLaci in Europe has decreased from 30 % in 1980, with areas of very high AAE over central Europe, to 2 % in 2010 with an expected decrease to just 1 % exceeded area in 2030 (Slootweg et al., 2014). The exceeded area of CL for eutrophication (CLeutN) in Europe in 1980 has



been calculated to 75 %, decreasing to 65 % in 2010 and projected to still be quite substantial in 2030 with 49 % exceeded

area (Slootweg et al., 2014).

While land-based emissions have decreased significantly as a result of international agreements and abatement measures, the regulations implemented for the shipping sector have been more modest. Emissions generated from international shipping are significant in areas with heavy marine traffic, and the Baltic Sea is one of the most highly trafficked seas in the world and

shipping is projected to increase in the future (Baltic LINes 2016).

International shipping is regulated by the International Maritime Organization (IMO), a body of the United Nations. Environmental pollution from ships is regulated by IMO's International Convention on the Prevention of Pollution from Ships (MARPOL 73/78) and its annexes. The MARPOL Annex VI - "Regulations for the Prevention of Air Pollution from Ships" -

sets limits on emissions of SOx and NOx from international shipping.

Regulation of emissions of S are through maximum allowed fuel S content. Until recently, the S limit in fuel globally was 3.5% (mass to mass). This has been reduced to 0.5% by 1st of January 2020. Annex VI enables the establishment of Emission Control Areas (ECAs) with more stringent emission limits both for S (SECA) and for NOx (NECA). The Baltic Sea has been,

together with the North Sea and the English Channel, established as SECA in 2006. The fuel S limit that applied in SECA until 1st of January 2015 was 1%, after that date it was decreased to 0.1%. Installation of exhaust-gas aftertreatment equipment, cleaning the exhaust gas to concentrations corresponding to those at use of a clean fuel, is allowed as an alternative. In the EU the S regulations in Annex VI are implemented through the EU sulphur directive which has additional restrictions: the limit for passenger ships in regular traffic outside of the SECA area was 1.5% until 2020, and for ships at berth in EU ports the limit

has been set to 0.1 % since 2010.

IMO's NOx regulation sets emission limits through Tiers which apply limiting curves depending on maximum engine operating speed according to the year when the ship was built. The Tier I and Tier II apply to ships built between 2000 and 2011 and after 2011, respectively. While the Tier I limit has in practice not any effect the Tier II limit reduces the NOx

emissions by 16-20 %. The Tier III decreases the emissions by 80% and requires exhaust gas cleaning equipment (selective catalytic reduction or exhaust gas recirculation) or alternative fuel, such as liquefied natural- or biogas (LNG, LBG). The Tier III controls apply only to the specified ships while operating in NECAs established to limit NOx emissions, outside such areas the Tier II controls apply. The NECA that includes the Baltic Sea, the North Sea and the English Channel has gone into effect 2021 and applies to ships built after January 1st 2021.


On average, 70 % of shipping emissions are released within a distance of 400 km from the coast and can have significant air quality impacts on coastal regions (e.g. Corbett et al., 1999; Eyring et al., 2005; Sofiev et al., 2018; Jonson et al., 2020). It is



expected that the relative contribution from international shipping to air pollutant emissions will increase, as land-based
reductions in emissions occur at a faster rate.


In this paper we investigate the role of shipping emissions of S and N in the exceedances of critical loads for the area around
the Baltic Sea for the year 2012 and for a number of emission scenarios for the year 2040.

## 2 Material and methods

The deposition of S and N due to the various scenarios has been developed within the Sustainable Shipping and Environment
of the Baltic Sea region (BONUS SHEBA) project, whose purpose has been to take a holistic approach to emissions from
shipping and its impact on the environment in a number of important areas, including that of SOx and NOx emissions on land
ecosystems of the Baltic Sea region. By combining the deposition from the current and in the project developed shipping
scenarios, calculated with help of a chemistry transport model, with the critical load data used under the LRTAP Convention,
the exceedances of CLs for acidity and eutrophication have been evaluated for the Baltic Sea area. The model set-up and
scenarios are briefly described in the next sections, but for a more detailed description and evaluation of the model performance
the reader is referred to Karl et al. (2019a, b).

### 2.1 Model setup

The deposition of S and N has been simulated using the regional atmospheric chemistry transport CMAQ (Community
Multiscale Air Quality) model (Byun and Schere, 2006). The CMAQ model simulations were driven by the meteorological
fields of the COSMO-CLM version 5.0 (Rockel et al., 2008) using the ERA-Interim re-analysis (Dee et al., 2011) as forcing
data for the year 2012. 2012 was assessed to be a good reference year for the Baltic Sea area according to an analysis of the
temperature anomalies and precipitation anomalies for the decade 2004–2014 for the Baltic proper (Karl et al., 2019a). The
2012 meteorology was also used for the 2040 scenarios to allow direct comparison between the simulations.

Land-based emissions have been calculated using the emission model SMOKE for Europe (SMOKE-EU, Bieser 2011) version
2.4. These emissions are based on national totals from EMEP (www.ceip.at), further details are given in Bieser (2011) and
Karl et al. (2019a). Emissions for the scenarios in 2040 are based on changes in emissions of the air pollutants in question
between 2010 and 2040 in the Baltic Sea countries, differentiated for different emission sectors according to ECLIPSE v.5
'Current Legislation' (CLE) 'Base' scenario (Amann et al., 2014).


Shipping emissions in the Baltic Sea and the North Sea were calculated using the STEAM model (Ship Traffic Emission
Assessment Model, Jalkanen et al 2009; 2012; Johansson et al 2013). These are based on position data from individual ships
collected from AIS (Automatic Identification System) data and include all merchant ships larger than 300 GT. The model





calculates emissions of $SO_x$, $NO_x$, $CO_2$, CO and particulate matter, differentiated in its components $SO_4^{2-}$, mineral ash and
elementary and organic carbon, in high temporal and spatial resolution.

The model was run in 3 nested domains with resolutions 64 km × 64 km for the whole continent, 16 km × 16 km for Central
and Northern Europe and 4 km × 4 km for the Baltic Sea region. The results from the domain with a resolution of 16 km × 16
km were used for the calculations for exceedances of the CLs. Results from hemispheric modelling with the SILAM model
(Sofiev et al., 2015) on a resolution of 0.5° × 0.5° were used as boundary conditions for the chemical fields in the simulations.

The model output has a time resolution considerably higher (hourly) than what is needed for the exceedance calculations
(yearly). The deposition data have been aggregated to yearly sums of N and S for the respective scenario. The model works
with a chemical mechanism including the most common S and N species present (Table 1) and both wet and dry deposition
are included in the calculations.

Several of the modelled species are of major importance for atmospheric chemical reactions but are not quantitatively important
for the deposition. The largest proportions of the total deposition of N are from $HNO_3$, $NH_3$, particulate $NH_4^+$ and $NO_3^-$
(together approximately 95% of total annual deposition). The S deposition is mostly from $SO_2$ and particulate $SO_4^{2-}$. Karl et
al. (2019a) compared the modelled wet deposition of oxidized and reduced N for 2012 with data from EMEP monitoring
stations and found an underestimation in the CMAQ simulations for all stations in the Baltic Sea Region by 20-80%. While
for Germany the reason could be underestimation of $NH_3$ emissions from agriculture, comparison of modelled and measured
$NH_3$ concentrations in Denmark and Poland shows overestimation by the model, indicating that the reason for underestimation
of N deposition in these areas is rather availability of sulphuric and nitric acid or limited formation of particulate ammonium
nitrate and sulphate.

**2.2 Shipping scenario description**

The years 2012 and 2040 were chosen to evaluate the impact of shipping on the Baltic Sea region. 2012 was before the
introduction of the more stringent SECA limit of 0.1% fuel S content in the Baltic Sea and 2040 was chosen in the BONUS
SHEBA project as the scenario time horizon representing the future when the current already agreed legislation, if fully
implemented, will have full impact and when we still can assume non-disruptive development of shipping technology. To
investigate impacts of NECA, to be introduced in 2021, there needs to be enough time for a substantial part of the fleet to be
renewed, since it will only apply to ships built after the 1st of January 2021.

The baseline scenario for 2040 developed in the BONUS SHEBA project is the so-called Business As Usual (BAU) scenario
that takes into account trends of economic growth and development of shipping, extrapolating the current trends, as well as
the predefined regulations. An important parameter in the future scenarios is development of energy efficiency in shipping,





which is regulated by IMO regulation on Energy Efficiency Design Index (EEDI). The BAU scenario adopts development of the energy efficiency in shipping according to Kalli et al. (2013), assuming the annual efficiency increases of 1.3% to 2.25 %, depending on ship type, which is beyond the IMO EEDI regulation (corresponding efficiency increase values 0.65% to 1.04

%) and significantly reduces shipping fuel consumption. Even though higher than the EEDI regulation, the energy effectivization in this scenario is still not sufficient to meet ─ in extrapolation ─ the IMO goal to reduce fossil $CO_2$ emissions from shipping by 50% until 2050. Regarding other predefined legislation, the BAU scenario assumes implementation and full compliance with the NECA regulation in the Baltic Sea, North Sea and the English Channel in 2021 (Tier III for all ships built (keel laid) 2021 and later, operating in the region), implementation and full compliance with the SECA 0.1% fuel S content

limit in this region in 2015 and implementation of the global 0.5% fuel S content limit in 2020. More details about this scenario can be found in Karl et al., 2019a. In order to investigate impact of NECA and of the energy effectivization in the shipping sector two sensitivity scenarios were investigated: one without implementation of NECA (BAU-NoNECA) and another with effectivization that just follows the IMO EEDI regulation (EEDI) (see Table 2).

**2.3 Critical loads of acidity and eutrophication**

A critical load (CL) is defined as "a quantitative estimate of an exposure to one or more pollutants below which significant harmful effects on specified sensitive elements of the environment do not occur according to present knowledge" (Nilsson and Grennfelt, 1988). Critical loads are calculated for different receptors (e.g., terrestrial ecosystems, aquatic ecosystems), and a sensitive element can be any part (or the whole) of an ecosystem or ecosystem process. It is up to each country to decide what ecosystem or what sensitive part of an ecosystem they use as a receptor in CL calculations. The area for which the CL is

calculated (EcoArea) is then the sensitive area of a country which needs to be protected from air pollution. Critical loads have been defined to avoid the eutrophying effects of N deposition (critical loads of eutrophying N, CLeutN) and for the acidifying effects of both S and N deposition (critical loads of acidity).

The critical load for a site is mostly calculated from steady-state (charge and) mass balance equation(s), linking a chemical

criterion (e.g., an acceptable N concentration in soil solution or a critical ANC limit in lake water) with the corresponding deposition value(s). For eutrophication CLs, also empirical values have been defined for many habitat types (Bobbink and Hettelingh, 2011). While the eutrophication CL is a single number (to be compared with N deposition), the acidity critical load is defined as a trapezoidal function (critical load function, CLF) in the (Ndep, Sdep) plane, defined by three characteristic numbers: CLmaxS, CLminN and CLmaxN. Methods to compute CLs are summarised in the Mapping Manual (UNECE, 2004;

see also Posch et al., 2015), which is used under the LRTAP Convention.

If a deposition is higher than the critical load at a site, the CL is said to be exceeded. For eutrophication, the exceedance is defined as the difference between N deposition (Ndep) and the CL, i.e. Ex = Ndep – CLeutN (and set to zero if negative). In the case of acidity both N and S deposition have to be set in relation to the CLF: it is defined as Ex = ExN + ExS, where ExN





and ExS are the amounts of Sdep and Ndep to be reduced to reach the point on the CLF that is closest to (Ndep, Sdep). For technical details and calculation procedures see Posch et al. (2015). For reporting and mapping purposes a single exceedance number is computed for each grid cell (or any other region). This number, called the average accumulated exceedance (AAE), is defined as the weighted mean of all ecosystems within the grid cell, with the weights being the respective ecosystem areas (see Posch et al., 2001). Note, that the ecosystem areas, for which CLs for acidity and eutrophication are determined, can differ

in some countries/regions since the ecosystems that are to be protected from acidification (e,g, lakes) may not be the same as ecosystems threatened by eutrophication (e.g. Natura 2000 areas).

Exceedances in this paper are calculated using the critical load database held at the Coordination Centre for Effects under the LRTAP Convention (Hettelingh et al., 2017) and used in supporting European assessments and negotiations on emission

reductions (e.g., Reis et al., 2012; EEA Report, 2014). Critical loads of acidity and of eutrophication were calculated for 72% of the modelled area (13 042 grid cells on a 0.15˚ x 0.15˚ grid). The remaining cells are either sea areas or modelled grid cells covering land areas not relevant for critical load calculations, such as agricultural land or human settlements. CL exceedance calculations for individual counties in Sweden were performed with the same methodology and based on the same critical loads database.

**2 Results**

**2.1 Atmospheric deposition of S**

At small geographical scale, the modelled deposition of S originating from shipping emissions is highest in the shipping lanes and in the coastal areas (Figure 1, lower panels). At the geographical area of the whole (extended) Baltic Sea region, the overall gradient is from high deposition near large land-based emission sources in the south and south-east towards low deposition in

the north and north-west (Figure 1, upper panels.). There is a strong decrease in S deposition between 2012 and 2040 (Figure 1, Table 3) due to decreased emissions from both land-based sources and shipping. The deposition from shipping emissions is expected to decrease by approximately 90% from 2012 to 2040 (Figure 2, Table 3). In 2040 the differences between the shipping scenarios are marginal, since the decisive legislation, i.e., the introduction of SECA in the Baltic Sea, has been implemented in all future scenarios. Any differences in the S deposition between the 2040 BAU and the 2040 BAU-NoNECA

scenarios are caused by the differences in atmospheric processes regarding oxidation of $SO_2$ as well as formation and subsequent deposition of ammonium sulphate. By 2040 the S deposition caused by the shipping emissions is not expected to exceed 0.2 kg ha$^{-1}$ yr$^{-1}$ anywhere, not even in the shipping lanes (Figure 1, lower right panel).

**2.2 Atmospheric deposition of N**

Similar to the deposition of S, there is a geographical gradient in N deposition from high deposition in the south to low

deposition in the north. This pattern roughly follows the density of agriculture and of other land-based N emissions sources,





such as traffic. The overall decrease in N deposition from 2012 to 2040 (Figure 2) is due to declining emissions from land-based sources and from shipping. The change is pronounced at the whole modelled geographical domain, but the decrease is less strong than that of S. There is a clear decrease in N deposition originating from shipping between 2012 and 2040 for all scenarios (Figure 3). However, in 2040 there is still a significant amount of N deposition originating from shipping and the

differences between the individual scenarios in 2040 are large. The implementation of NECA has major impact on N deposition from shipping in most of the coastal areas in the Baltic region (Figure 3), with the exception of the Gulf of Bothnia in the north, where the shipping intensity is low. Without introducing a NECA (scenario BAU-NoNECA) the contribution to N deposition would in median be more than twice as big as in the BAU case (Scenario BAU, Table 3). While the implementation of a NECA has a large impact for the whole region, the lower energy-effectivisation scenario (EEDI) makes the biggest

difference on the west coast of Sweden. The areas with the highest N deposition originating from shipping emissions overlap with the part of the region with highest exceedance of critical loads for eutrophication (see below, Figure 6).

**2.3 Contribution of shipping to the deposition of S and N**

In the year 2012 deposition of S was still relatively high, reaching to >5 kg ha$^{-1}$ yr$^{-1}$ at the 1% of most impacted parts of the modelled area (Table 3) of which shipping in the Baltic region contributed by >0.8 kg ha$^{-1}$ yr$^{-1}$. In the year 2040, the deposition

of S has decreased massively and the contribution from Baltic shipping is in absolute terms very low, above 0.09 kg ha$^{-1}$ yr$^{-1}$ only at the most impacted 1% of the modelled area (Table 3, Figure 4). The land areas with high deposition of S typically receive high deposition originating both, from Baltic shipping and from other sources (Figure 1). Consequently, the Baltic shipping emissions of S and to an even larger extent of N (Figure 2) impact predominantly the parts of the modelled domain already under pressure from other air pollution sources.


As stated above, the change from 2012 to 2040 (Table 3, Figures 1, 2 and 3) followed a similar pattern for both S and N, but the decrease to the year 2040 has been less strong for N in all modelled scenarios. Cumulative distribution of the depositions in 2012 and 2040 for different scenarios (Figure 4) provides further details. Non-implementation of a NECA would have caused Baltic shipping N emissions contribute to deposition on average more than twice as much than in the BAU scenario,

and the less stringent demands on energy effectivization of the EEDI scenario would have a clear and quantitatively important impact in terms of N deposition (Figure 4, Table 3).

**2.4 Critical load exceedances of CLaci**

The extent and magnitude of the exceedances of CLs for acidification in 2012 and 2040 is presented in Figure 5. The AAE (average accumulated exceedance) is a measure of how much the deposition exceeds what the affected areas can withstand

(on average in a grid cell; Posch et al., 2015). Geographically the areas with CLaci exceedances are concentrated at the Swedish west coast and in northern Germany. The different scenarios for the year 2040 differ only in shipping, while the other factors, such as meteorology and emissions from other sources, are the same for these scenarios.





The impact of shipping on CL exceedance can be seen by comparing the NoShip-scenario maps on the right with the total

scenario maps in the left-hand panels in Figure 5. For acidification the impact is limited, except for some areas with relatively

higher AAE on the Swedish west coast, in northern Germany and in southern Lithuania.

There is an improvement (decrease) in the exceedances of acidification in the 2040 scenarios compared to 2012. This is,

however, also due to emission reductions of the land-based sources. When comparing the two lower panels of Figure 5 there

are only minimal, hardly visible differences, indicating that there is only a very small impact from the Baltic Sea and North

Sea shipping in 2040 with regards to the exceedances for acidification in 2040.

All the investigated scenarios include the same regulation for S in marine fuels and there is not much difference between the

scenarios. Neither the introduction of a NECA nor the lower energy efficiency impact the exceedances for acidification to any

larger extent.

CL exceedances are calculated and presented on a national scale, in each country for the ecosystem area in need of protection

(EcoArea). Tables 4 and 5 present the exceedances of CLaci (for both EcoArea with CL exceedance and as AAE) for all

countries surrounding the Baltic Sea. Of the 10 countries, the total country area is modelled for five countries (EE, FI, LT, LV

and SE; see Figure 1 for the country-codes) while the modelled domain extends only partially over the remaining five countries

(DE, DK, NO, PL and RU). Caution must be exercised when interpreting the calculated exceedance for countries not entirely

covered by the modelled area considered here, since any exceedance percentage is not the percentage for their respective total

EcoArea.

The countries most affected by exceedances of CLaci in 2012 are Lithuania and Sweden, both for area and AAE (Tables 4 and

5). The modelled northern part of Germany has the highest exceeded area and AAE, but that is only a small part of the whole

country and an interpretation for the whole country is therefore not meaningful, based on these results alone. Comparing the

results for 2012 with the 2012 NoShip scenario, it shows that shipping has some impact on the exceedances. There is a slight

area increase and, especially for the northern part of Germany, quite a substantial increase in AAE. For the rest of the countries

the resulting exceedances are low, both for area and AAE. The results for the 2040 scenarios show several countries with very

low or zero exceedances. For the countries with the largest exceedances in 2012 there is a great improvement, and the impact

of shipping in the year 2040 is rather insignificant.



## 2.5 Critical load exceedances of CLeutN

Eutrophication is clearly a problem, both in terms of area with exceedance and for AAE values, for most of the land ecosystems around the Baltic Sea, except for the north and north-eastern part of the Baltic coast (Figure 6). The largest CLeutN exceedances are in Denmark and on the west coast of Sweden. The impact of shipping on CL exceedances occurs in most of the coastal areas of the central and southern Baltic Sea, including the entire southern Sweden.

For the exceedances of CLs for eutrophication there are clear improvements seen in the 2040 scenarios (Figure 7), although eutrophication is still a problem over large parts of the area. There is still a contribution from shipping to the exceedances, although it is less pronounced than in 2012 (cf Figure 6). Without the introduction of NECA (Figure 7, lower left panel), the exceedances will be higher, especially in Denmark and in the southwest and south of Sweden.

Tables 4 and 5 present the exceedances of CLeutN. The exceedances of CLs for eutrophication in 2012 are far greater than those for acidification. Lithuania, Latvia and Estonia each have a large percentage of exceeded area, ranging from almost 50% to over 90% (Tables 6 and 7). The situation in Denmark (small parts of the west coast not included in the modelled area) is arguably the worst with almost the whole area exceeded (Table 6) and very high AAE. The northern part of Germany also has high exceedances of CLs for eutrophication. Comparisons of the results for the 2012 and the 2012 NoShip scenarios

demonstrate the contribution that shipping has to the exceedances for eutrophication and thereby how much reduced shipping emissions could help to alleviate the situation. In some countries, shipping emissions increase both the exceeded area and the AAE (e.g., Latvia and Estonia). In others, e.g., Denmark and Lithuania, where the area exceeded is high, there is a smaller further increase in area exceeded, but a substantial increase in AAE because the shipping emissions impact areas where the CL are already exceeded.


For the year 2040 there is a marked decrease in the CLeutN exceedances for all countries, although several still have high exceedances (highest at DK, DE and LV). Comparing the year 2040 NoShip scenario with the other scenarios, shipping is contributing to the exceedances although to a lesser extent than in 2012. Of the three 2040 scenarios where shipping is included, the BAU scenario with the higher energy efficiency and the introduction of NECA has the lowest exceedances. The highest

exceedances are found in the scenario where the NECA is not introduced (BAU_NoNECA). The EEDI-scenario which has implemented NECA but with lower energy efficiency lies in-between the other two scenarios but is closer to the BAU scenario.

## 2.6 The impact of emissions from shipping on Sweden

Given the large geographical extent of the area impacted by the emissions from the Baltic Sea shipping, the division of the impacts by countries (Tables 5 and 6) is logical but hides the fact that the effects are often unevenly distributed within each



country. To illustrate the variability of the impact of shipping emissions within a country we have chosen to look at Sweden
        in more detail.

        Sweden, a country of 450 000 km$^2$ and with an extension of almost 1 600 km from south to north, has a large geographical
        difference in the exceedance of the CLs. In general, the CL exceedance gradient is from the Swedish south-west coast which
is most affected by both eutrophication and acidification, to the northern part of the country where there is, and has been,
        relatively less deposition of S and N and lowest CL exceedances. Importantly for the contribution from the Baltic Sea shipping,
        Sweden has several large ports with heavy shipping traffic and areas close to major shipping routes. The Port of Gothenburg,
        located on the Swedish west coast, is the largest port in Scandinavia. Other include the Port of Trelleborg in the south of
        Sweden, port of Helsingborg and Malmö, both by the Öresund Straight on the west coast and the Port of Stockholm on the
east coast. Taking a closer look at the impact of shipping for Sweden, the exceedances of CLs for acidification and
        eutrophication have been calculated on a county level for the scenarios in Table 2.

        Table 8 gives a list of the Swedish counties and the average deposition of N and S. In 2012, Hallands län (county *N*) on the
        Swedish west coast received the highest total deposition of S (2.6 kg ha$^{-1}$ yr$^{-1}$, calculated as an average for the EcoArea in the
county) followed by counties *I* and *M* in the south and south east (2.4 kg ha$^{-1}$ yr$^{-1}$) (see also Figure 1). The northernmost
        counties receive quite little deposition in comparison (0.5-1.0 kg ha$^{-1}$ yr$^{-1}$). The highest deposition of S originating from
        shipping is also over county *N*, closely followed by county *AB* (both around 0.5 kg ha$^{-1}$ yr$^{-1}$) and then *O*, *M* and *I* (close to 0.4
        kg ha$^{-1}$ yr$^{-1}$). Counties *AB* and *O* receive the largest contribution from shipping in relative terms (21% and 19%, respectively)

The highest average deposition of N (on acidification-sensitive ecosystems) is on the south and southwest of Sweden. County
        *N* receives the highest deposition (9.1 kg ha$^{-1}$ yr$^{-1}$) followed by counties *M* and *K* (8.2 and 7.3 kg ha$^{-1}$ yr$^{-1}$). The least amount
        of N deposition is in northern Sweden (1-3 kg ha$^{-1}$ yr$^{-1}$, Table 8). The contribution of N deposition from shipping is largest
        over county *N* as well (1.5 kg ha$^{-1}$ yr$^{-1}$), followed by county *K* and *O* (1.3 and 1.2 kg ha$^{-1}$ yr$^{-1}$). In relative terms, the largest
        contribution from shipping is in counties *W*, *S* and *H* (close to 20%).


        The area of exceedance for the CLs for acidification in Sweden in 2012 is close to 5% (Table 9). Even though the total
        percentage is low, there are areas with high exceedances. County *N* has the highest exceedances in both area and AAE for CLs
        for acidification (30.4%, 40.2 eq ha$^{-1}$ yr$^{-1}$). Shipping emissions (2012 – 2012-NoShip) contribute here with 7 % of the exceeded
        EcoArea and the AAE is doubled. Counties *I* and *M* have little to no exceedance when it comes to acidification, even though
they receive a similar amount of S deposition as county *N*. This is due to the low sensitivity to acidification in these regions
        (county *I* is rich in limestone e.g.). On the other hand, counties *G* and *S* have the second and third highest exceedances of CL
        for acidification.





The area of exceedance for CL for eutrophication in Sweden for 2012 is almost three times as much as for acidification (about
14%). Several of the counties have over 90% of their areas exceeded with respect to eutrophication, many of which receive a
relatively high input of deposition from shipping emissions (Figure 3). Although the difference in exceeded area for these
counties is relatively small between the 2012 and the 2012-NoShip scenario, the input from shipping is reflected in the
increased AAE. County $N$ has the largest AAE for eutrophication and is exceeded on 99.9 % of the ecosystem areas sensitive
for eutrophication. The difference in AAE between the 2012 and the 2012-NoShip scenario for county $N$ is close to 300 eq ha$^{-1}$
$y^{-1}$.

The contribution from shipping to the total deposition of S was between 8% and 21% at the county level in 2012. Until 2040,
it is expected to decrease to between 2% to 4% (Table S3). The decrease in S deposition can be mainly attributed to harder
restrictions such as the introduction of a SECA (0.1%) in 2015, but other developments such as higher fuel efficiency, changes
in fuel type, economic development etc also have an impact.

The contribution from shipping to the total deposition of N was between 14% and 19% at the county level in 2012. Until 2040,
it is expected to decrease to between 5% and 7% (Table 10).

The area of the exceedance of CLs for acidification drops from 5% in 2012 to about 2% for the BAU 2040 scenario. Generally,
there is very little difference in the exceedances of CLs for acidification between any of the 2040 scenarios, including the
2040-NoShip scenario. There seems to be little impact of NECA and lower fuel efficiency and even of shipping in general on
the critical load exceedances for acidity.

The area of the exceedance of CLs for eutrophication drops from 14% in 2012 to about half in the BAU scenario for 2040. In
contrast to the results for acidification there is a larger difference between the 2040 scenarios. In the scenario where the NECA
has not been implemented (BAU-NoNECA), the exceeded area is, as expected, greater than for BAU and EEDI. The scenario
with the NECA implemented, but with the lower energy efficiency (EEDI scenario) has slightly higher exceedances than BAU
but is lower than BAU-NoNECA. In contrast to exceedances of CLs for acidification, shipping still has an impact on
exceedances for eutrophication in 2040 (Tables 11 and 12). For several of the counties (e.g., $O$, $S$, $K$, $F$, $I$) there is a clear
improvement with the implementation of the NECA in the exceedances of CLs for eutrophication.

## 3 Discussion and conclusions

The introduction of SECA and NECA for the Baltic Sea and the North Sea, together with increased energy efficiency of ship
operations and other changes included in the future scenarios showed a reduction of S and N deposition from shipping even
though transport volumes are expected to increase. A decrease of S deposition originating from land-based sources from 2012





to 2040 by ca 53% in median for the modelled extended Baltic Sea area was calculated. The implementation of SECA resulted in even stronger relative decrease of shipping-related deposition of S, contributing in 2040 in all 3 scenarios in median less than 1% of the remaining total S deposition. For N, the total deposition (land-based plus from shipping) decreased by ca 43% (median). The contribution from shipping decreased from ca 10% in 2012 to 4 - 9% of the total in 2040, depending on the

modelled scenario. The less strong deposition decline for N and the less strong decrease of the shipping contribution explains the differences in exceedances of CLaci and CLeutN.

Jonson et al. (2019) investigated the effects of the Baltic Sea ECA regulations by comparing, among others, the oxidized N and S deposition between 2014, 2016 and 2030. They found that while shipping in the Baltic Sea was contributing with more

than 10% to S deposition before strengthening of the fuel S limit in 2015, it became an insignificant source of S deposition in 2016, after implementation of the 0.1 % fuel S limit, and further decrease of S deposition between 2016 and 2030 is mainly due to changes in land-based emissions. Jonson et al. (2019) expected a reduction in the contribution of Baltic Sea shipping to the deposition of oxidized N by 40-50% between 2016 and 2030. In our results we can see an even larger decrease from 2012 to 2040, part of which can be explained by more ships being affected by the NECA rules.


The maps with shipping-related deposition of oxidised S (Fig. 1, bottom panels) and oxidised N (Fig. 3) show a distinct difference in regional pattern, S deposition having maxima along the shipping lines while the highest deposition of oxidised N is along the coasts. The main reason for the high deposition of S along the shipping lanes is that $SO_2$ is highly water soluble, especially in alkaline sea water, causing the emitted $SO_2$ to quickly deposit on the water surface near the source. The process

is further enhanced by rainwater washing out the $SO_2$. The shipping plumes are emitted at low altitude and the boundary layer above the sea is mostly neutral. Therefore, the exchange between the plumes and the water surface is efficient. On land, the largest contribution from shipping to the total S deposition is along the coasts, where some $SO_2$ has already been oxidized and the deposition consists of both $SO_2$ and particle-bound sulphate. The modelled deposition of the shipping-related N has a different pattern from that of S, with a higher deposition over the land than over the sea near the shipping lanes (Figure 3). The

reason is most likely that the deposition rate of $NO_2$ is low and the N deposition is completely dominated by the deposition of N in higher oxidized states. Most of the emitted $NO_2$ must first undergo oxidation in the atmosphere before it is deposited, primarily as particulate nitrate, some also as $HNO_3$ or organic nitrates. The nitrates are also less soluble in deliquescent aerosol particles and rain droplets than sulphate, especially if these are acidified. The N deposition in coastal areas may be enhanced due to more efficient deposition of particles over land than on the sea surface. Furthermore, the $HNO_3$ formed during transport

of maritime emissions over the sea can react in coastal areas with $NH_3$ from agriculture and form additional particulate nitrates. The pattern of N deposition from shipping is similar to the deposition pattern shown in Jonson et al. (2015) of modelled N deposition originating from shipping using the EMEP model.





The exceedances of CLs for both, acidity and eutrophication, decrease from 2012 to 2040 in all scenarios. There are substantial
reductions in land-based emissions that lead to decreases in the exceedances for acidity and eutrophication. For CLaci, when
comparing the 2040 scenarios, it shows that the impact of the NECA and of the lower fuel efficiency does not have any
noticeable impact on the critical load exceedances. Overall, the contribution from shipping is very low since the S emissions
are drastically reduced by the introduction of SECA, which is common to all scenarios.

By 2040, there will still be a significant area with exceedance of CLs for eutrophication with a noticeable contribution from
shipping. The highest impacts are on Denmark and northern Germany. Although the estimated exceedance of CLs for
eutrophication for the whole of Sweden in 2040 does not show a significant impact from shipping, this looks quite different
on a county level. The introduction of NECA will improve the situation in several of the Swedish counties, but more reductions
might be necessary to further reduce the impact of shipping, there. Repka et al. (2021) calculated the costs and environmental
benefits of ship-originated $SO_x$ and $NO_x$ emission reductions in the Baltic Sea and also assessed the effect of shipping emissions
on the exceedances of critical loads. While the underlying critical loads database (Hettelingh et al., 2017) used by Repka et al.
is the same as in this work, the geographical domain of the assessment, the models used to calculate atmospheric deposition
and the years of the assessment differ. Nevertheless, the overall conclusion about the relatively larger importance of shipping
emissions for exceedances of critical loads for eutrophication as opposed to acidification and about the decline in CL
exceedances from 2015 to 2030 (from 2012 to 2040 in our work) agree. Direct comparisons between the calculated CL
exceedances are difficult due to the different years considered. However, with that caveat, relatively large differences in the
calculated critical loads exceedance were found for Lithuania and for Latvia, while results were very similar for e.g. Finland
and Sweden. The differences in CL exceedances are driven by the differences in calculated atmospheric deposition by the
different models, which is in detail addressed by Karl et al., (2019b). The comparison of the two works provides an estimate
of uncertainty of the calculations due to e.g. different model set-ups and different meteorology between the modelled years,
combined with the change of CL exceedances over time, however, without providing an opportunity to separate the two.

The NECA rules that have been introduced in 2021 only apply to newly built vessels. With a life expectancy of approximately
25 years for ships, significant effects will be seen only about 15 years after the introduction. A national legislation that would
445    speed up the installation of $NO_x$ cleaning technology on ships, like the $NO_x$ fund in Norway, has a great potential to reduce
emissions far earlier, probably already in the next decade (Parsmo et al., 2017). The possibility of significantly reducing CL
exceedances through further measures in the shipping sector should be utilized even though other sectors of course also will
need to be involved.

450    An aspect to be considered is the recent IMO target of halving the $CO_2$ shipping emissions by 2050 compared to the 2008
emissions. The decision was made in April 2018 and is not included in the future scenarios in this paper. The reduction in
fossil fuel use that will be required to achieve this goal is more far-reaching than what has been adopted in the BAU scenario.





For all the future scenarios in this paper there is an assumption of 100% compliance with regards to all regulations. An investigation of S emissions from ships in Danish waters after the SECA entered into force showed a compliance rate between 92-97% depending on time period and platform used, the fuel S content of the non-compliant fuel varied between 0.3 % and 1.5 % (Mellqvist et al., 2017). This rate of noncompliance would increase the $SO_2$ emissions in order of 20-80%, though from the very low level. Compliance monitoring and efficient enforcement of especially NECA regulation, but also of fuel S content regulations are currently subjects of intensive research and discussion, not only by environmentalists but also by lawyers and policy-makers.

*Data availability*

The Critical loads database is currently hosted by Coordination Centre for Effects (CCE) at Umwelt Bundesamt in Germany (www.umweltbundesamt.de/en/Coordination_Centre_for_Effects)

*Code availability*

The air quality model CMAQ is developed and maintained by the U.S. Environmental Protection Agency (US EPA). COSMO-CLM is the community model of the German climate research (https://www.clm-community.eu/, last access: 26 February 2021). The simulations with COSMO-CLM and CMAQ were performed at the German Climate Computing Centre (DKRZ) within the project "Regional Atmospheric Modelling" (project ID 0302).

*Author contributions*

FM, SJ and JM conceived the paper. MK and VM modelled deposition from shipping, MP calculated CL exceedance and produce CL exceedance maps. SJ, FM and JM wrote the manuscript with ideas and feedback from all co-authors.

*Competing interests*

Authors declare that they have no competing interests.

*Acknowledgements*

*Financial support.* This work is part of the BONUS SHEBA (Sustainable Shipping and Environment of the Baltic Sea region) research project under Call 2014-41. BONUS (Art 185) is funded jointly by the EU and the national governments of the participating partners countries.

The national funds were provided by the German Federal Ministry of Education and Research under grant number 03F0720A (VM and MK), and the Swedish Environmental Protection Agency (SJ, FM and JM). The Swedish Environmental Protection Agency provided also funds through project NV-07751-17 "Sjöfartens utsläpp och kritisk belastning". The work on the manuscript (SJ, FM, JM) was also supported by the project platform CSHIPP, which has received funding from subsidy





contract C006 of Interreg Baltic Sea Region. MP thanks for earlier support by the Trust Fund for the effect-oriented activities under the LRTAP Convention.

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





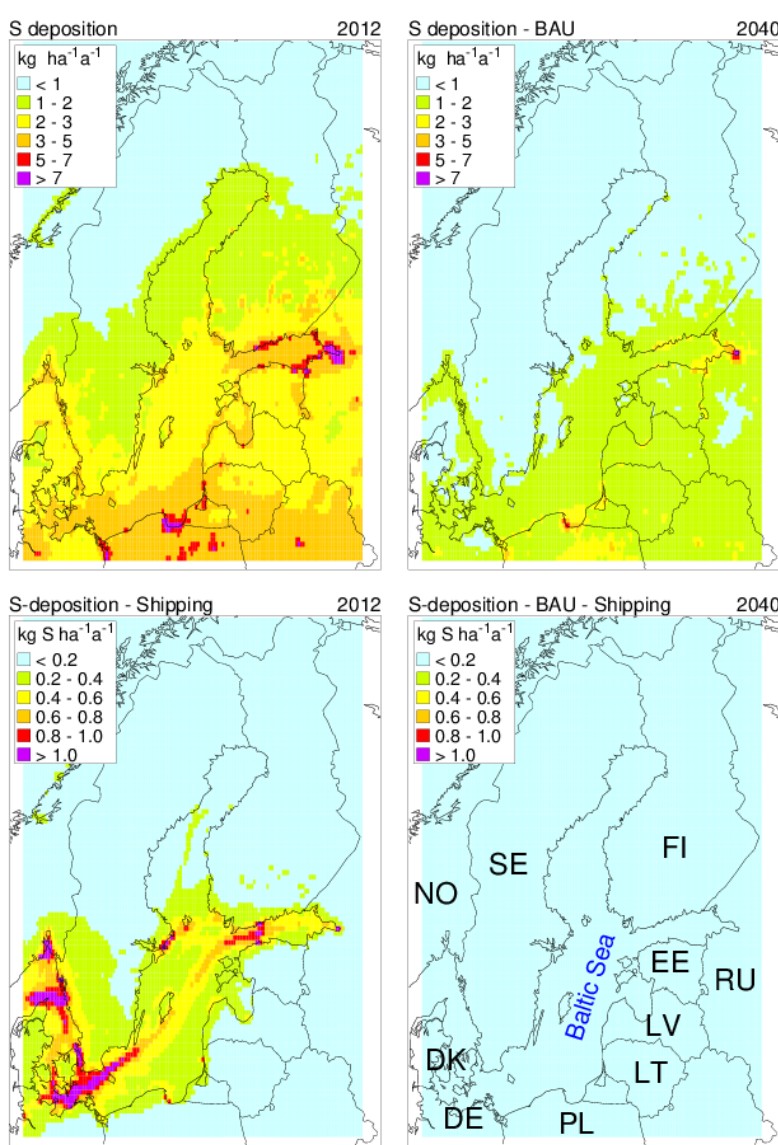

**Figure 1. Top: Total deposition of S in 2012 (left) and in 2040 under the BAU scenario (right) in the extended Baltic sea region. Bottom: S deposition originating from shipping in 2012 (left) and in 2040 (right), calculated as the difference between the scenario based on total emissions and a scenario where shipping emissions are removed. In the bottom-right map the countries are labelled with their ISO-3166 2-letter codes.**



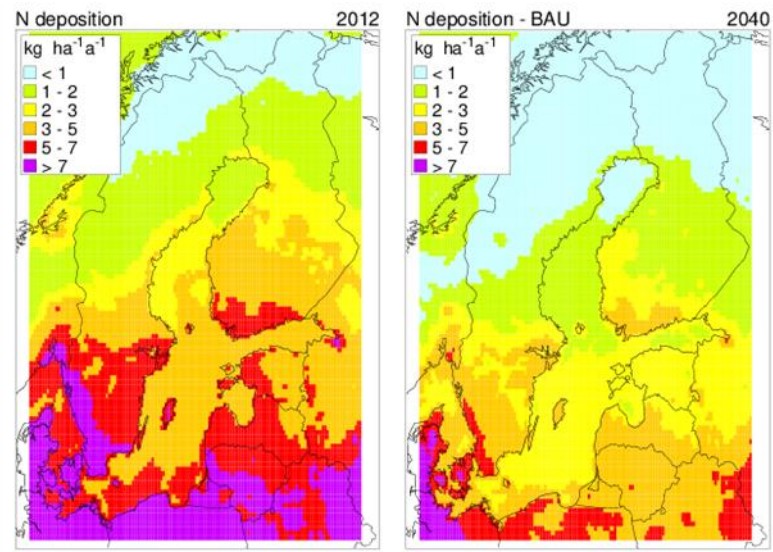

**Figure 2. Total deposition of N deposition in 2012 (left) and the 2040-BAU scenario (right).**





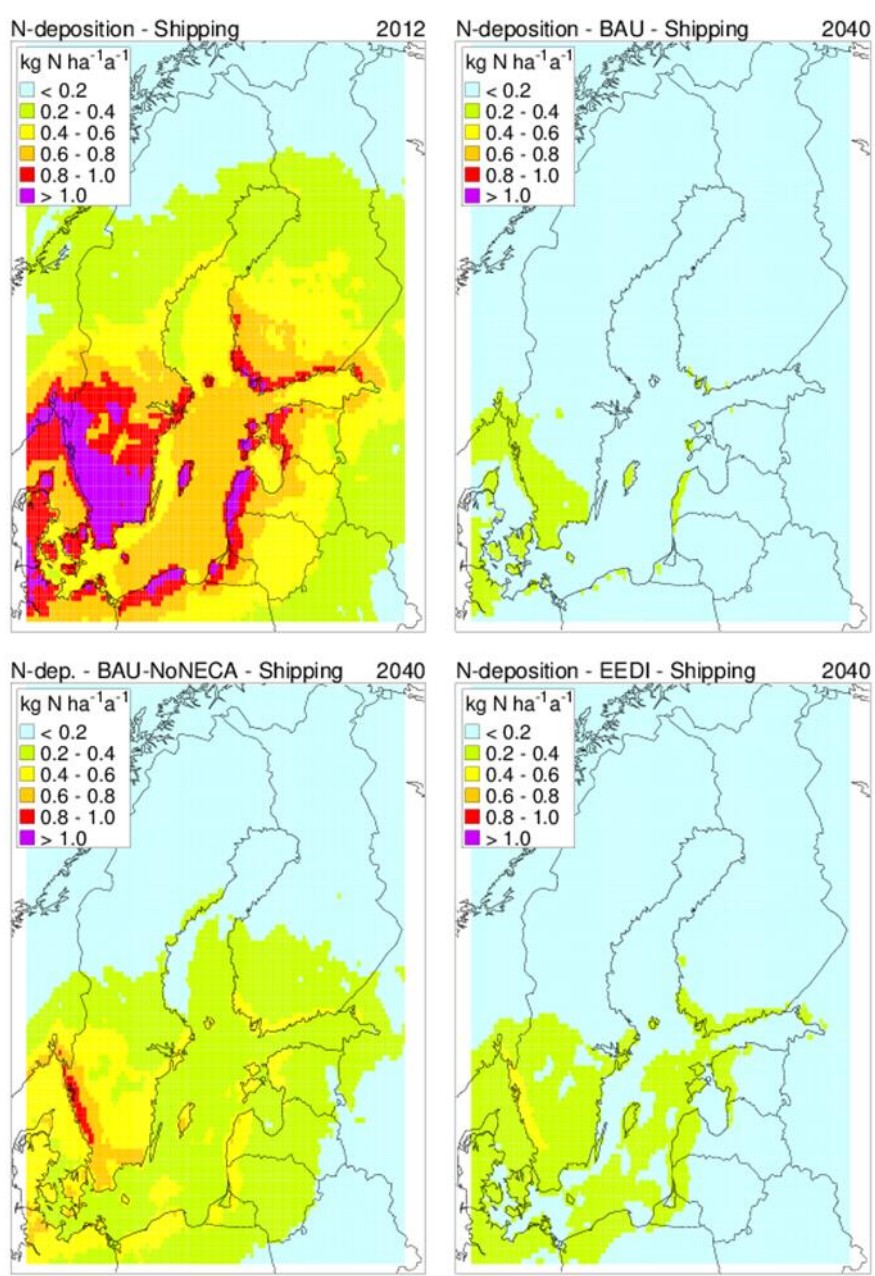

**Figure 3. N deposition originating from shipping in 2012 (top left) and in 2040 (BAU top right, BAU-NoNECA bottom left, EEDI bottom right). Deposition from shipping was calculated as the difference between the scenario based on total emissions and the scenario where shipping has been removed.**






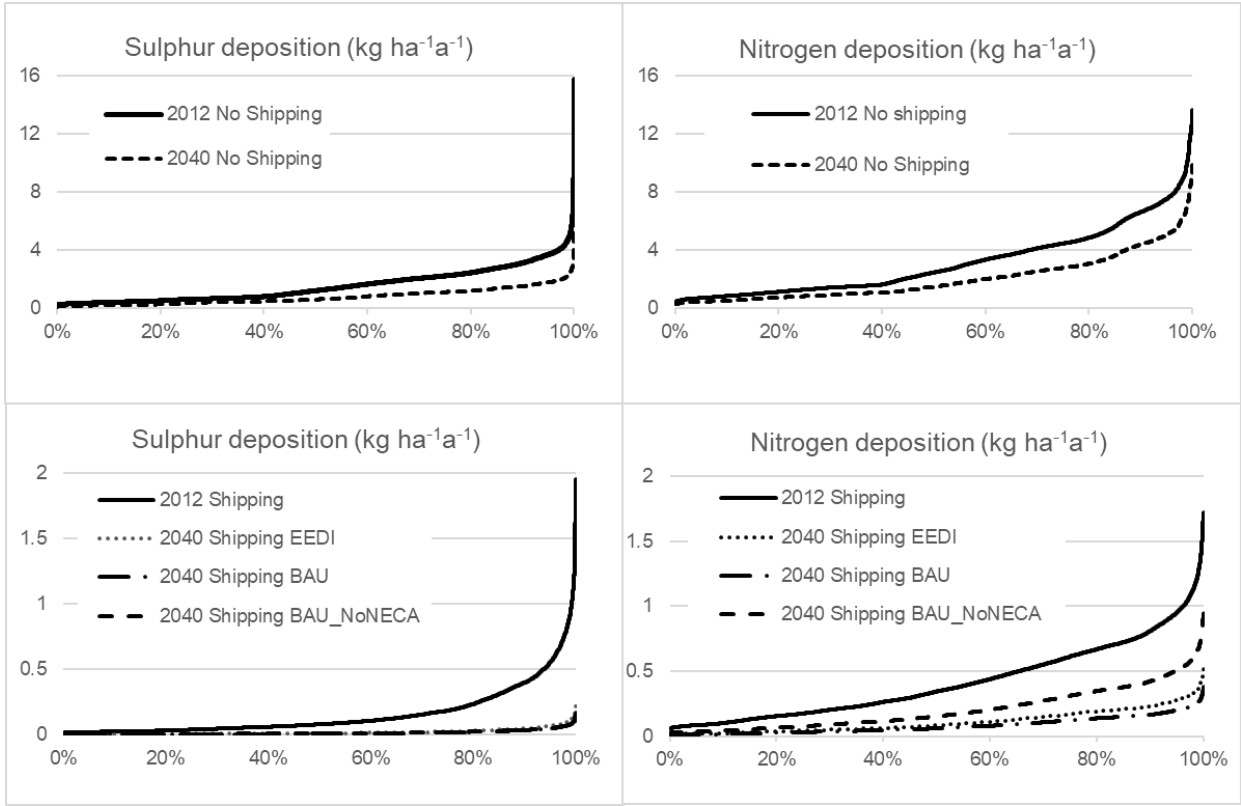

**Figure 4. Cumulative distribution of deposition of S (left hand panels) and N (right hand panels) for all grid cells in the modelled area. Top panels are the deposition from other sources (land-based, shipping from outside the North Sea and Baltic Sea), bottom panels deposition from Baltic shipping only. Note the different scales on the y-axis between the top and the bottom panels.**






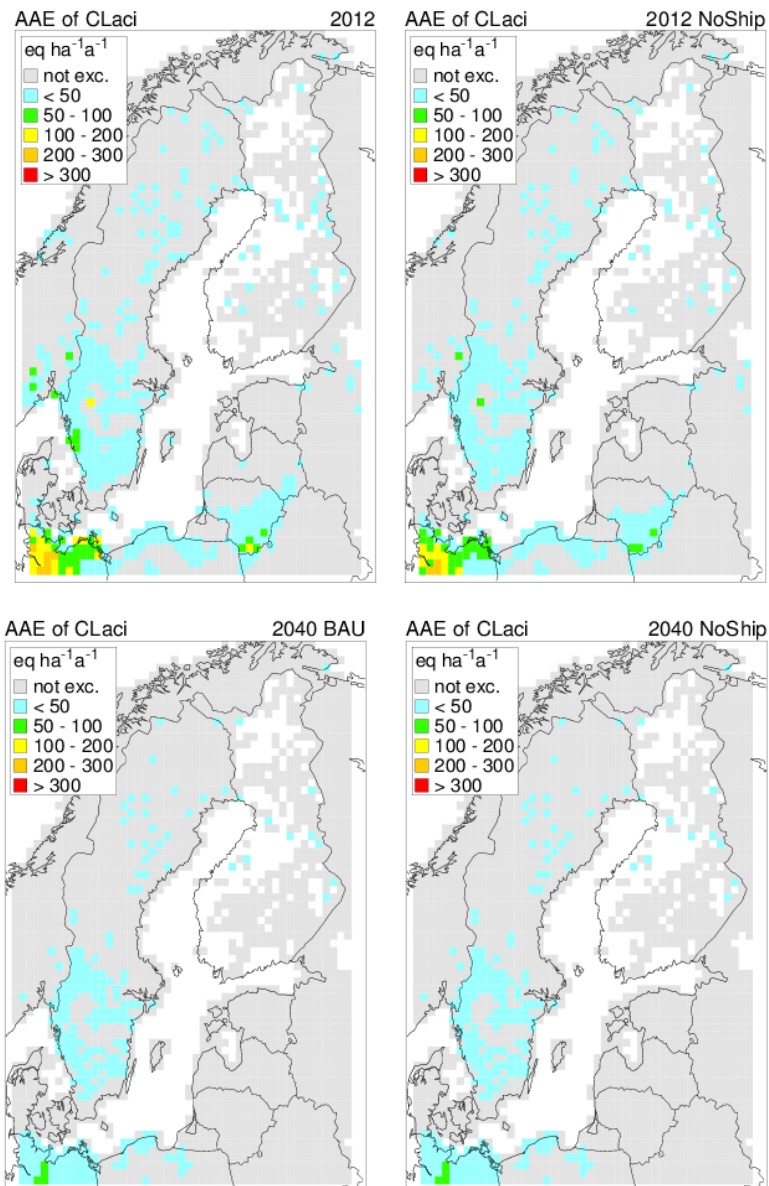

**Figure 5. The average accumulated exceedance (AAE) of CLaci 2012 (top) and 2040 (bottom). The left panels show the exceedances for the total deposition. The right panels show the exceedances for the scenario without shipping deposition. The differences in the 2040 NoShip and BAU scenarios for CLaci (lower panels) are very small, the differences in 2040 for CLaci between BAU, BAU-NoNeca and EEDI are negligible (maps not shown).**




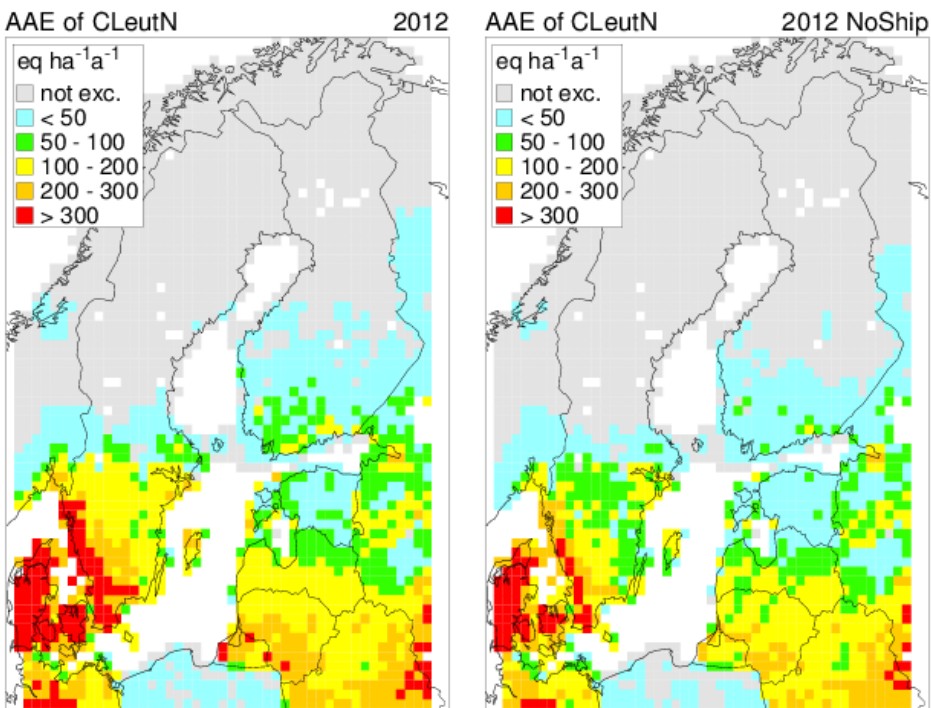

**Figure 6. The average accumulated exceedance (AAE) of CLeutN in 2012. The left panel show the exceedances for the total deposition. The right panel show the exceedances for the scenario without shipping deposition.**





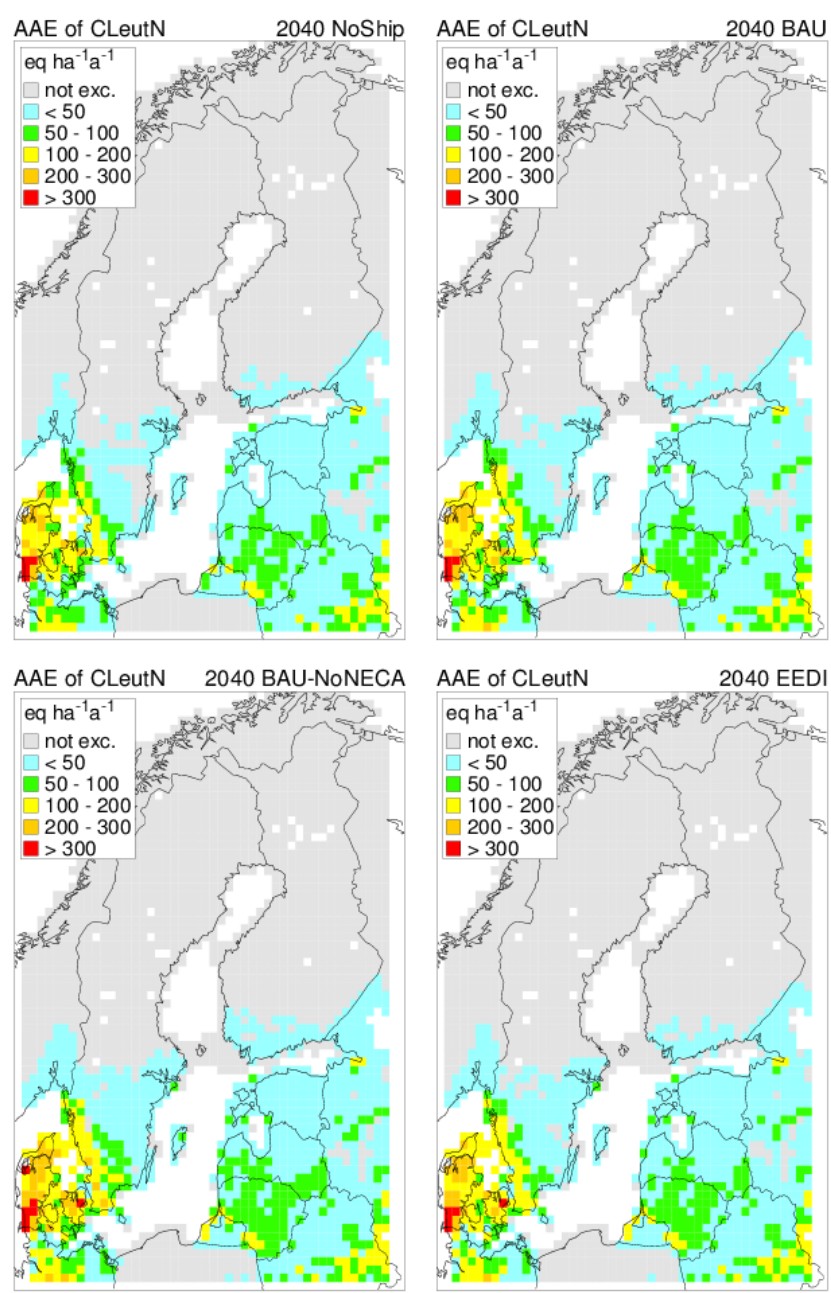


**Figure 7. The average accumulated exceedance (AAE) of CLeutN in 2040 for the four different scenarios.**



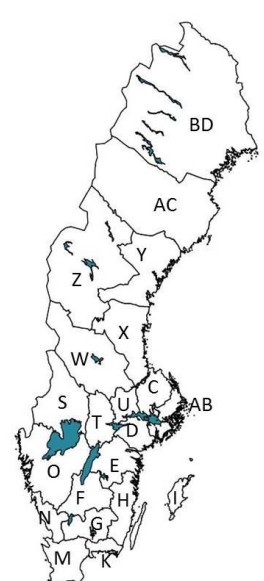

**Figure 8. Map of Sweden with the county borders and county letter-codes. County names can be seen in Table 8.**






**Table 1. S and N species in CMAQ deposition listed approximately in order of importance, species in italics contribute with less than 1%.**

| N species: | particulate $NO_3^-$, particulate $NH_4^+$, $HNO_3$, $NH_3$, $NO_2$, PAN, $N_2O_5$, *$HNO_4$, HONO, NO, OPAN* |
|---|---|
| S species: | particulate $SO_4^{2-}$, $SO_2$, *$H_2SO_4$* |

**Table 2. Description of the evaluated scenarios, including emission scaling factors that give emissions for the 2040 scenarios in relation to 2012 emissions (from Karl et al., 2019a).**

| Scenario | Description | Scaling factors shipping emissions | Scaling factors land-based emissions |
|---|---|---|---|
| **2012** | | | |
| **2012 NoShip** | no shipping included in the CMAQ model domains (included in the model boundary conditions) | | |
| **2040 BAU** | 'Business as usual' | $SO_x$: 0.088<br>$NO_x$: 0.207 | $SO_2$: 0.45<br>$NO_x$: 0.40<br>$NH_3$: 0.80 |
| **2040 BAU-NoNECA** | The same as 2040 BAU scenario without the implementation of NECA | $SO_x$: 0.088<br>$NO_x$: 0.505 | $SO_2$: 0.45<br>$NO_x$: 0.40<br>$NH_3$: 0.80 |
| **2040 EEDI** | This scenario has a lower energy efficiency than the 2040 BAU scenario, NECA is implemented | $SO_x$: 0.207<br>$NO_x$: 0.285 | $SO_2$: 0.45<br>$NO_x$: 0.40<br>$NH_3$: 0.80 |
| **2040 NoShip:** | no shipping included in the CMAQ model domains (included in the model boundary conditions) | | $SO_2$: 0.45<br>$NO_x$: 0.40<br>$NH_3$: 0.80 |





**Table 3. S and N deposition originating from the land-based sources and from shipping in 2012 and for the three scenarios in 2040. Minimum, maximum and percentiles were calculated on 13,042 grid cells with calculated critical loads for acidity and/or for eutrophication.**

| | S | kg ha⁻¹ yr⁻¹ | | | | |
|---|---|---|---|---|---|---|
| **Year** | 2012 | 2012 | 2040 | 2040 | 2040 | 2040 |
| | Land-based sources | Shipping | Land-based sources | Shipping BAU | Shipping BAU-No NECA | Shipping EEDI |
| **Min** | 0.2 | 0.0 | 0.1 | 0.00 | 0.00 | 0.00 |
| **1%** | 0.3 | 0.0 | 0.2 | 0.00 | 0.00 | 0.00 |
| **5%** | 0.3 | 0.0 | 0.2 | 0.00 | 0.00 | 0.00 |
| **50%** | 1.2 | 0.1 | 0.6 | 0.01 | 0.01 | 0.01 |
| **95%** | 3.8 | 0.4 | 1.8 | 0.04 | 0.04 | 0.05 |
| **99%** | 5.1 | 0.8 | 2.4 | 0.07 | 0.07 | 0.09 |
| **Max** | 15.7 | 1.9 | 7.3 | 0.17 | 0.17 | 0.23 |

| | N | kg ha⁻¹ yr⁻¹ | | | | |
|---|---|---|---|---|---|---|
| **Year** | 2012 | 2012 | 2040 | 2040 | 2040 | 2040 |
| | Land-based sources | Shipping | Land-based sources | Shipping BAU | Shipping BAU-No NECA | Shipping EEDI |
| **Min** | 0.4 | 0.1 | 0.3 | 0.01 | 0.02 | 0.01 |
| **1%** | 0.5 | 0.1 | 0.3 | 0.01 | 0.03 | 0.01 |
| **5%** | 0.7 | 0.1 | 0.4 | 0.01 | 0.03 | 0.02 |
| **50%** | 2.6 | 0.3 | 1.6 | 0.06 | 0.15 | 0.08 |
| **95%** | 7.9 | 1.0 | 5.3 | 0.21 | 0.53 | 0.29 |
| **99%** | 10.7 | 1.3 | 7.5 | 0.27 | 0.70 | 0.38 |
| **Max** | 13.7 | 1.7 | 9.8 | 0.38 | 0.97 | 0.53 |




**Table 4. Exceeded area (in % of EcoArea) of CLs for acidity for each country included in the modelled area for all scenarios.**

| Country | EcoArea (km²) | Exceeded area (%) CLs for acidity | | | | | |
|---|---|---|---|---|---|---|---|
| | | 2012_NoShip | 2012 | 2040_NoShip | 2040_BAU | 2040_BAU_NoNECA | 2040_EEDI |
| EE | 27232 | 0.1 | 0.1 | 0.0 | 0.0 | 0.0 | 0.0 |
| FI | 286 | 0.5 | 0.7 | 0.3 | 0.3 | 0.3 | 0.3 |
| LT | 22198 | 18.1 | 21.6 | 0.0 | 0.0 | 0.0 | 0.0 |
| LV | 36631 | 0.0 | 0.8 | 0.0 | 0.0 | 0.0 | 0.0 |
| SE | 395225 | 4.0 | 4.9 | 2.1 | 2.1 | 2.2 | 2.2 |
| DE* | 6917 | 28.8 | 31.6 | 14.1 | 14.5 | 15.5 | 14.9 |
| DK* | 4283 | 0.4 | 0.5 | 0.0 | 0.0 | 0.0 | 0.0 |
| NO* | 223218 | 1.2 | 2.4 | 0.2 | 0.2 | 0.2 | 0.2 |
| PL* | 26376 | 1.1 | 1.3 | 0.2 | 0.3 | 0.3 | 0.3 |
| RU* | 114120 | 0.0 | 0.1 | 0.0 | 0.0 | 0.0 | 0.0 |

* These countries do not have their entire area included in the modelled area.

**Table 5. Average Accumulated Exceedance (eq ha⁻¹ yr⁻¹) of CLs for acidity for each country included in the modelled area for all scenarios.**

| Country | EcoArea (km²) | AAE (eq ha⁻¹ yr⁻¹) CLs for acidity | | | | | |
|---|---|---|---|---|---|---|---|
| | | 2012_NoShip | 2012 | 2040_NoShip | 2040_BAU | 2040_BAU_NoNECA | 2040_EEDI |
| EE | 27232 | 0.1 | 0.1 | 0.0 | 0.0 | 0.0 | 0.0 |
| FI | 286 | 0.3 | 0.4 | 0.1 | 0.1 | 0.1 | 0.1 |
| LT | 22198 | 12.0 | 19.4 | 0.0 | 0.0 | 0.0 | 0.0 |
| LV | 36631 | 0.0 | 0.1 | 0.0 | 0.0 | 0.0 | 0.0 |
| SE | 395225 | 2.1 | 3.2 | 0.6 | 0.7 | 0.7 | 0.7 |
| DE* | 6917 | 83.1 | 103.0 | 20.9 | 22.5 | 24.9 | 23.2 |
| DK* | 4283 | 0.2 | 0.6 | 0.0 | 0.0 | 0.0 | 0.0 |
| NO* | 223218 | 0.6 | 1.6 | 0.0 | 0.1 | 0.1 | 0.1 |
| PL* | 26376 | 1.1 | 1.5 | 0.1 | 0.1 | 0.1 | 0.1 |
| RU* | 114120 | 0.0 | 0.0 | 0.0 | 0.0 | 0.0 | 0.0 |




**Table 6. Exceeded area (in % of EcoArea) of CLs for eutrophication for each country included in the modelled area for all scenarios**

| Country | EcoArea (km²) | Exceeded area (%) CLs for eutrophication | | | | | |
|---|---|---|---|---|---|---|---|
| | | 2012_NoShip | 2012 | 2040_NoShip | 2040_BAU | 2040_BAU_NoNECA | 2040_EEDI |
| EE | 27232 | 24.6 | 48.6 | 9.1 | 9.5 | 10.5 | 9.6 |
| FI | 41140 | 3.7 | 7.7 | 0.1 | 0.2 | 0.3 | 0.2 |
| LT | 22198 | 88.2 | 93.9 | 40.7 | 42.4 | 45.2 | 43.1 |
| LV | 36631 | 57.3 | 79.0 | 35.4 | 35.6 | 36.5 | 35.7 |
| SE | 58688 | 13.6 | 14.4 | 5.9 | 6.9 | 9.5 | 7.6 |
| | | | | | | | |
| DE* | 6917 | 74.2 | 81.8 | 39.6 | 41.0 | 43.2 | 41.4 |
| DK* | 4283 | 96.0 | 98.9 | 76.4 | 78.7 | 82.7 | 79.7 |
| NO* | 211133 | 4.3 | 6.1 | 0.4 | 0.6 | 1.1 | 0.8 |
| PL* | 26376 | 0.6 | 1.4 | 0.0 | 0.0 | 0.0 | 0.0 |
| RU* | 114120 | 39.4 | 46.0 | 11.7 | 12.3 | 13.3 | 12.5 |

**Table 7. Average Accumulated Exceedance (eq ha⁻¹ yr⁻¹) of CLs for eutrophication for each country included in the modelled area for all scenarios.**

| Country | EcoArea (km²) | AAE (eq ha⁻¹ yr⁻¹) CLs for eutrophication | | | | | |
|---|---|---|---|---|---|---|---|
| | | 2012_NoShip | 2012 | 2040_NoShip | 2040_BAU | 2040_BAU_NoNECA | 2040_EEDI |
| EE | 27232 | 24.9 | 42.3 | 6.5 | 7.3 | 8.6 | 7.6 |
| FI | 41140 | 1.2 | 3.4 | 0.0 | 0.0 | 0.1 | 0.0 |
| LT | 22198 | 164.4 | 200.2 | 54.1 | 57.0 | 61.9 | 58.2 |
| LV | 36631 | 90.0 | 121.0 | 35.0 | 37.9 | 42.5 | 39.0 |
| SE | 58688 | 17.7 | 26.3 | 3.2 | 4.1 | 5.9 | 4.5 |
| | | | | | | | |
| DE* | 6917 | 191.7 | 233.3 | 71.0 | 75.9 | 83.7 | 77.8 |
| DK* | 4283 | 327.3 | 388.6 | 156.8 | 168.1 | 186.2 | 172.6 |
| NO* | 211133 | 3.2 | 6.9 | 0.2 | 0.2 | 0.5 | 0.3 |
| PL* | 26376 | 0.3 | 0.7 | 0.0 | 0.0 | 0.0 | 0.0 |
| RU* | 114120 | 37.6 | 51.3 | 8.1 | 8.8 | 9.8 | 9.0 |



**Table 8. Average deposition of N and S (kg ha⁻¹ yr⁻¹) in Swedish counties for 2012 and 2012 NoShip.**

| County | | km² EcoArea | 2012 NoShip kg ha$^{-1}$ yr$^{-1}$ N | kg ha$^{-1}$ yr$^{-1}$ S | 2012 kg ha$^{-1}$ yr$^{-1}$ N | kg ha$^{-1}$ yr$^{-1}$ S |
|---|---|---|---|---|---|---|
| AB | Stockholms län | 4055 | 4.5 | 1.8 | 5.4 | 2.2 |
| C | Uppsala län | 6241 | 3.9 | 1.3 | 4.5 | 1.5 |
| D | Södermanlands län | 5507 | 4.8 | 1.6 | 5.6 | 1.8 |
| E | Östergötlands län | 9306 | 4.7 | 1.5 | 5.6 | 1.6 |
| F | Jönköpings län | 8809 | 5.2 | 1.6 | 6.3 | 1.8 |
| G | Kronobergs län | 9238 | 5.7 | 1.9 | 6.9 | 2.2 |
| H | Kalmar län | 7941 | 4.7 | 1.7 | 5.7 | 1.9 |
| I | Gotlands län | 1603 | 5.3 | 2.0 | 6.3 | 2.4 |
| K | Blekinge län | 2564 | 6.1 | 1.9 | 7.3 | 2.2 |
| M | Skåne län | 6043 | 7.0 | 2.0 | 8.2 | 2.4 |
| N | Hallands län | 4399 | 7.6 | 2.2 | 9.1 | 2.6 |
| O | Västra Götalands län | 16804 | 5.9 | 1.7 | 7.2 | 2.1 |
| S | Värmlands län | 18395 | 3.9 | 1.3 | 4.8 | 1.5 |
| T | Örebro län | 8842 | 4.1 | 1.3 | 5.0 | 1.5 |
| U | Västmanlands län | 3729 | 3.9 | 1.2 | 4.6 | 1.4 |
| W | Dalarnas län | 29176 | 2.4 | 0.8 | 3.0 | 1.0 |
| X | Gävleborgs län | 17437 | 2.4 | 0.9 | 2.9 | 1.0 |
| Y | Västernorrlands län | 24039 | 1.8 | 0.7 | 2.2 | 0.8 |
| Z | Jämtlands län | 50016 | 1.3 | 0.4 | 1.6 | 0.5 |
| AC | Västerbottens län | 59297 | 1.2 | 0.5 | 1.4 | 0.6 |
| BD | Norrbottens län | 101787 | 0.8 | 0.4 | 1.0 | 0.5 |
| SWEDEN | | 395226 | 2.4 | 0.8 | 2.9 | 1.0 |







**Table 9. Exceedances of CLs for acidification (aci) and eutrophication (eut) for both area (%) and AAE (eq ha$^{-1}$ yr$^{-1}$) for 2012 and 2012-NoShip for the respective EcoAreas (EcoAraci=EcoArea for acidification, EcoAreut=EcoArea for eutrophication).**

| | | 2012-NoShip | | 2012 | | | 2012-NoShip | | 2012 | |
| | | | eq ha$^{-1}$ | | eq ha$^{-1}$ | | | eq ha$^{-1}$ | | eq ha$^{-1}$ |
| | km2 | % | yr$^{-1}$ | % | yr$^{-1}$ | km2 | % | yr$^{-1}$ | % | yr$^{-1}$ |
| County | EcoAraci | Ex%aci | AAEaci | Ex%aci | AAEaci | EcoAreut | Ex%eut | AAEeut | Ex%eut | AAEeut |
|---|---|---|---|---|---|---|---|---|---|---|
| AB | 4055 | 1.5 | 1.4 | 1.5 | 1.8 | 179 | 86.3 | 104.3 | 93.4 | 155.9 |
| C | 6241 | 0.0 | 0.0 | 0.0 | 0.0 | 526 | 60.8 | 26.9 | 75.0 | 58.7 |
| D | 5507 | 10.4 | 6.3 | 11.0 | 7.6 | 447 | 96.0 | 120.9 | 99.8 | 171.4 |
| E | 9306 | 6.0 | 3.8 | 6.6 | 4.5 | 1273 | 95.8 | 76.4 | 99.4 | 123.7 |
| F | 8809 | 7.5 | 4.0 | 8.9 | 5.5 | 842 | 98.9 | 129.1 | 100.0 | 193.3 |
| G | 9238 | 19.2 | 11.5 | 23.9 | 16.3 | 204 | 98.7 | 188.3 | 100.0 | 271.7 |
| H | 7941 | 9.6 | 5.1 | 11.5 | 6.7 | 829 | 91.1 | 96.4 | 96.2 | 147.0 |
| I | 1603 | 0.0 | 0.0 | 1.3 | 0.4 | 186 | 94.2 | 90.0 | 99.1 | 149.6 |
| K | 2564 | 0.0 | 0.0 | 0.0 | 0.0 | 252 | 97.1 | 147.3 | 97.1 | 210.5 |
| M | 6043 | 3.0 | 2.8 | 3.6 | 3.5 | 881 | 94.8 | 218.0 | 95.4 | 276.4 |
| N | 4399 | 23.1 | 19.8 | 30.4 | 40.2 | 290 | 99.9 | 300.9 | 99.9 | 399.6 |
| O | 16804 | 17.5 | 11.8 | 21.4 | 20.7 | 1465 | 98.1 | 152.3 | 98.3 | 225.2 |
| S | 18395 | 21.4 | 9.2 | 26.8 | 12.6 | 882 | 83.2 | 56.8 | 90.0 | 109.4 |
| T | 8842 | 13.3 | 7.1 | 16.3 | 9.0 | 235 | 84.5 | 65.4 | 97.0 | 120.1 |
| U | 3729 | 7.6 | 2.1 | 10.0 | 3.0 | 116 | 55.3 | 53.2 | 59.5 | 78.1 |
| W | 29176 | 2.0 | 0.5 | 2.2 | 0.7 | 2552 | 1.6 | 0.3 | 5.5 | 2.2 |
| X | 17437 | 1.0 | 0.1 | 1.3 | 0.2 | 303 | 7.2 | 0.9 | 25.8 | 7.6 |
| Y | 24039 | 0.9 | 0.2 | 1.0 | 0.3 | 372 | 0.0 | 0.0 | 0.0 | 0.0 |
| Z | 50016 | 0.0 | 0.0 | 0.1 | 0.0 | 6216 | 0.0 | 0.0 | 0.0 | 0.0 |
| AC | 59297 | 0.7 | 0.2 | 0.8 | 0.2 | 11152 | 0.0 | 0.0 | 0.2 | 0.0 |
| BD | 101787 | 0.4 | 0.1 | 0.5 | 0.1 | 29486 | 0.0 | 0.0 | 0.0 | 0.0 |
| Sweden | 395226 | 4.0 | 2.1 | 4.9 | 3.2 | 58688 | 13.6 | 17.7 | 14.4 | 26.3 |




**Table 10. Average deposition of N and S (kg ha⁻¹ yr⁻¹) in the Swedish counties for the 2040 scenarios (2040 NoShip, BAU, BAU-NoNECA, EEDI).**


| County | km² EcoArea | 2040 NoShip kg ha⁻¹ yr⁻¹ N | kg ha⁻¹ yr⁻¹ S | BAU kg ha⁻¹ yr⁻¹ N | eq ha⁻¹ yr⁻¹ S | BAU-NoNECA kg ha⁻¹ yr⁻¹ N | kg ha⁻¹ yr⁻¹ S | EEDI kg ha⁻¹ yr⁻¹ N | kg ha⁻¹ yr⁻¹ S |
|---|---|---|---|---|---|---|---|---|---|
| AB | 4055 | 2.9 | 0.9 | 3.1 | 0.9 | 3.3 | 0.9 | 3.1 | 0.9 |
| C | 6241 | 2.5 | 0.6 | 2.6 | 0.6 | 2.8 | 0.6 | 2.7 | 0.6 |
| D | 5507 | 3.1 | 0.8 | 3.2 | 0.8 | 3.5 | 0.8 | 3.3 | 0.8 |
| E | 9306 | 2.9 | 0.7 | 3.1 | 0.7 | 3.3 | 0.7 | 3.2 | 0.7 |
| F | 8809 | 3.2 | 0.7 | 3.4 | 0.8 | 3.7 | 0.8 | 3.5 | 0.8 |
| G | 9238 | 3.5 | 0.9 | 3.7 | 0.9 | 4.1 | 0.9 | 3.8 | 0.9 |
| H | 7941 | 2.8 | 0.8 | 3.0 | 0.8 | 3.3 | 0.8 | 3.1 | 0.8 |
| I | 1603 | 3.3 | 1.0 | 3.5 | 1.0 | 3.8 | 1.0 | 3.6 | 1.1 |
| K | 2564 | 3.6 | 0.9 | 3.9 | 0.9 | 4.3 | 1.0 | 4.0 | 1.0 |
| M | 6043 | 4.5 | 1.0 | 4.8 | 1.0 | 5.2 | 1.0 | 4.9 | 1.0 |
| N | 4399 | 4.8 | 1.1 | 5.1 | 1.2 | 5.6 | 1.2 | 5.2 | 1.2 |
| O | 16804 | 3.7 | 0.9 | 4.0 | 0.9 | 4.4 | 0.9 | 4.1 | 0.9 |
| S | 18395 | 2.3 | 0.7 | 2.5 | 0.7 | 2.8 | 0.7 | 2.6 | 0.7 |
| T | 8842 | 2.5 | 0.6 | 2.7 | 0.6 | 2.9 | 0.6 | 2.7 | 0.6 |
| U | 3729 | 2.4 | 0.6 | 2.6 | 0.6 | 2.8 | 0.6 | 2.6 | 0.6 |
| W | 29176 | 1.4 | 0.4 | 1.5 | 0.4 | 1.7 | 0.4 | 1.6 | 0.4 |
| X | 17437 | 1.4 | 0.4 | 1.5 | 0.5 | 1.6 | 0.5 | 1.5 | 0.5 |
| Y | 24039 | 1.1 | 0.4 | 1.1 | 0.4 | 1.2 | 0.4 | 1.2 | 0.4 |
| Z | 50016 | 0.8 | 0.2 | 0.9 | 0.2 | 0.9 | 0.2 | 0.9 | 0.2 |
| AC | 59297 | 0.7 | 0.3 | 0.8 | 0.3 | 0.8 | 0.3 | 0.8 | 0.3 |
| BD | 101787 | 0.5 | 0.2 | 0.5 | 0.2 | 0.6 | 0.2 | 0.5 | 0.2 |
| Sweden | 395226 | 1.5 | 0.4 | 1.6 | 0.4 | 1.7 | 0.4 | 1.6 | 0.4 |






**Table 11. Exceedances of CLs for acidification (aci) for area (%) and AAE (eq ha⁻¹ yr⁻¹) for the 2040 scenarios.**

| | | 2040-NoShip | | BAU | | BAU-NoNECA | | EEDI | |
|---|---|---|---|---|---|---|---|---|---|
| | km2 | % | eq ha⁻¹ yr⁻¹ | % | eq ha⁻¹ yr⁻¹ | % | eq ha⁻¹ yr⁻¹ | % | eq ha⁻¹ yr⁻¹ |
| County | EcoAraci | Ex%aci | AAEaci | Ex%aci | AAEaci | Ex%aci | AAEaci | Ex%aci | AAEaci |
| AB | 4055 | 0.6 | 0.5 | 0.6 | 0.6 | 0.6 | 0.6 | 0.6 | 0.6 |
| C | 6241 | 0.0 | 0.0 | 0.0 | 0.0 | 0.0 | 0.0 | 0.0 | 0.0 |
| D | 5507 | 5.8 | 2.2 | 5.8 | 2.3 | 5.8 | 2.3 | 5.8 | 2.3 |
| E | 9306 | 4.1 | 1.4 | 4.1 | 1.4 | 4.1 | 1.4 | 4.1 | 1.5 |
| F | 8809 | 3.4 | 1.2 | 3.4 | 1.2 | 3.7 | 1.2 | 3.7 | 1.2 |
| G | 9238 | 8.8 | 3.2 | 8.8 | 3.3 | 8.8 | 3.3 | 8.8 | 3.4 |
| H | 7941 | 4.7 | 1.3 | 5.1 | 1.3 | 5.1 | 1.3 | 5.1 | 1.3 |
| I | 1603 | 0.0 | 0.0 | 0.0 | 0.0 | 0.0 | 0.0 | 0.0 | 0.0 |
| K | 2564 | 0.0 | 0.0 | 0.0 | 0.0 | 0.0 | 0.0 | 0.0 | 0.0 |
| M | 6043 | 2.5 | 1.0 | 2.5 | 1.1 | 2.5 | 1.1 | 2.5 | 1.1 |
| N | 4399 | 11.9 | 5.1 | 11.9 | 5.4 | 11.9 | 5.4 | 11.9 | 5.5 |
| O | 16804 | 10.2 | 3.7 | 10.5 | 4.0 | 10.5 | 4.0 | 10.5 | 4.1 |
| S | 18395 | 11.1 | 2.8 | 11.3 | 2.9 | 11.3 | 2.9 | 11.3 | 3.0 |
| T | 8842 | 8.6 | 2.4 | 8.6 | 2.5 | 8.6 | 2.5 | 8.6 | 2.6 |
| U | 3729 | 1.9 | 0.4 | 1.9 | 0.4 | 1.9 | 0.4 | 1.9 | 0.4 |
| W | 29176 | 0.7 | 0.1 | 0.7 | 0.1 | 0.7 | 0.1 | 0.7 | 0.1 |
| X | 17437 | 0.0 | 0.0 | 0.0 | 0.0 | 0.0 | 0.0 | 0.0 | 0.0 |
| Y | 24039 | 0.6 | 0.1 | 0.6 | 0.1 | 0.6 | 0.1 | 0.6 | 0.1 |
| Z | 50016 | 0.0 | 0.0 | 0.0 | 0.0 | 0.0 | 0.0 | 0.0 | 0.0 |
| AC | 59297 | 0.5 | 0.1 | 0.5 | 0.1 | 0.5 | 0.1 | 0.5 | 0.1 |
| BD | 101787 | 0.2 | 0.0 | 0.2 | 0.0 | 0.2 | 0.0 | 0.2 | 0.0 |
| Sweden | 395226 | 2.1 | 0.6 | 2.1 | 0.7 | 2.2 | 0.7 | 2.2 | 0.7 |






**Table 12. Exceedances of CLs for eutrophication (eut) for both area (%) and AAE (eq**

**ha$^{-1}$ yr$^{-1}$) for the 2040 scenarios.**

| | | 2040-NoShip | | BAU | | BAU-NoNECA | | EEDI | |
|---|---|---|---|---|---|---|---|---|---|
| | km2 | % | eq ha$^{-1}$ yr$^{-1}$ | % | eq ha$^{-1}$ yr$^{-1}$ | % | eq ha$^{-1}$ yr$^{-1}$ | % | eq ha$^{-1}$ yr$^{-1}$ |
| County | EcoAreut | Ex%eut | AAEeut | Ex%eut | AAEeut | Ex%eut | AAEeut | Ex%eut | AAEeut |
| AB | 179 | 37.9 | 14.1 | 52.9 | 19.7 | 66.8 | 30.7 | 61.4 | 22.3 |
| C | 526 | 6.0 | 0.9 | 6.6 | 1.4 | 7.7 | 2.4 | 7.0 | 1.6 |
| D | 447 | 69.0 | 10.5 | 78.8 | 17.7 | 89.8 | 31.4 | 89.3 | 21.0 |
| E | 1273 | 7.0 | 1.1 | 10.0 | 2.0 | 46.4 | 7.1 | 21.7 | 2.7 |
| F | 842 | 51.5 | 11.1 | 77.1 | 19.0 | 96.1 | 36.7 | 84.8 | 23.1 |
| G | 204 | 87.2 | 42.9 | 87.6 | 57.0 | 87.7 | 79.7 | 87.6 | 62.7 |
| H | 829 | 38.0 | 7.3 | 40.9 | 11.8 | 61.8 | 21.6 | 48.1 | 13.9 |
| I | 186 | 17.1 | 3.7 | 34.5 | 7.3 | 65.7 | 17.1 | 39.4 | 9.3 |
| K | 252 | 56.9 | 20.4 | 63.8 | 29.9 | 86.8 | 45.7 | 66.1 | 33.9 |
| M | 881 | 87.6 | 76.6 | 87.9 | 89.3 | 89.3 | 108.9 | 87.9 | 94.2 |
| N | 290 | 91.7 | 105.8 | 94.3 | 126.0 | 97.1 | 157.7 | 94.3 | 134.2 |
| O | 1465 | 53.6 | 33.2 | 59.6 | 43.7 | 73.3 | 61.9 | 61.9 | 48.1 |
| S | 882 | 0.3 | 0.0 | 5.0 | 0.2 | 29.5 | 3.3 | 9.7 | 0.5 |
| T | 235 | 3.6 | 0.6 | 7.1 | 1.1 | 54.8 | 5.9 | 8.2 | 1.4 |
| U | 116 | 38.7 | 3.3 | 39.2 | 6.5 | 39.5 | 11.4 | 39.5 | 7.7 |
| W | 2552 | 0.0 | 0.0 | 0.0 | 0.0 | 0.0 | 0.0 | 0.0 | 0.0 |
| X | 303 | 0.0 | 0.0 | 0.0 | 0.0 | 0.0 | 0.0 | 0.0 | 0.0 |
| Y | 372 | 0.0 | 0.0 | 0.0 | 0.0 | 0.0 | 0.0 | 0.0 | 0.0 |
| Z | 6216 | 0.0 | 0.0 | 0.0 | 0.0 | 0.0 | 0.0 | 0.0 | 0.0 |
| AC | 11152 | 0.0 | 0.0 | 0.0 | 0.0 | 0.0 | 0.0 | 0.0 | 0.0 |
| BD | 29486 | 0.0 | 0.0 | 0.0 | 0.0 | 0.0 | 0.0 | 0.0 | 0.0 |
| Sweden | 58688 | 5.9 | 3.2 | 6.9 | 4.1 | 9.5 | 5.9 | 7.6 | 4.5 |