# Peer review of "The impact of nitrogen and sulphur emissions from shipping on exceedances of critical loads in the Baltic Sea region"

_Atmospheric Chemistry and Physics, 2021_

## Referee Comment (RC1)

Comments to

*The impact of nitrogen and sulphur emissions from shipping on exceedances of critical loads in the Baltic Sea region*
*By Sara Jutterström et al.*

Abstract
The abstract does not over the paper very well. Please add (at least) a sentence about the results of the sub-national analysis carried out for Sweden.

Line 39. *In the atmosphere ammonia reacts readily with both HNO$_3$ and H$_2$SO$_4$ forming particulate ammonium sulphate and nitrate ((NH$_4$)$_2$SO$_4$, NH$_4$NO$_3$).*
Change *HNO3 and H2SO4* to *H2SO4 and HNO3* to comply with the sequence used in the rest of the sentence and between the brackets.

Line 49. *but also eutrophication has decreased.*
This requires a reference

Line 59: *with the ultimate aim reducing depositions below the CLs*
Please change to 'with the ultimate aim of reducing depositions below CLs'

Line 79. *limits both for S (SECA) and for NOx (NECA).*
Inconsistent. Why not *S and N* or *SOx and NOx*

Line 128. *and include all merchant ships larger than 300 GT*
Please indicate what that means: how much of the total emissions is covered by limiting the data to ships of > 300GT?

Line 147-150. *While for Germany the reason could be underestimation of NH$_3$ emissions from agriculture, comparison of modelled and measured NH$_3$ concentrations in Denmark and Poland shows overestimation by the model, indicating that the reason for underestimation of N deposition in these areas is rather availability of sulphuric and nitric acid or limited formation of particulate ammonium nitrate and sulphate.*

This reasoning is hard to follow. Why would one assume an underestimation of NH3 emissions in Germany being the cause for underestimating N depositions while as at the same time the NH3 concentrations in the adjacent Denmark are overestimated by the model which may, i.m.o. indicate overestimation of emissions of NH3. Please rephrase.

Line 169. *Tier III for all ships built (keel laid) 2021 and later, operating in the region*
What does that addition mean, keel laid. Not clear. Is it needed, or is simply *built 2021 and later* enough?

Line 171. *In order to investigate impact of NECA*
Change to 'In order to investigate the impact of NECA'

Line 234: *Without introducing a NECA (scenario BAU-NoNECA) the contribution to N deposition would in median be more than twice as big as in the BAU case*
Unclear sentence. '*the contribution to N deposition would in median be more than twice as big*'. Which median is meant here? The median of the depositions computed over all grid cells? Please clarify.

Line 238: *In the year 2012 deposition of S was still relatively high, reaching to >5 kg ha$_{-1}$ yr$_{-1}$ at the 1% of most impacted parts of the modelled area*
How is *the most impacted part* defined? As the area with the highest total deposition or the area with the highest deposition originating from shipping emissions or.... Please explain.

Line 286: *For the countries with the largest exceedances in 2012 there is a great improvement, and the impact of shipping in the year 2040 is rather insignificant*

'Great improvement ' and ' insignificant' So the improvement is from other emission reductions I assume? Please explain.

Line 340. *The highest average deposition of N (on acidification-sensitive ecosystems)*
*acidification-sensitive ecosystems*: how have these been defined? Do you assume all ecosystems for which CL's are submitted by Sweden are sensitive, or did you make a selection based on e.g. CLs ? Please explain.

Line 428: *The introduction of NECA will improve the situation in several of the Swedish counties, but more reductions might be necessary to further reduce the impact of shipping, there.*
This is a vague statement. *More reduction **might be needed**.* Make this more conclusive: what **is** needed in light of the AAE and how much can reducing shipping emissions contribute to that.

Line 452. *The reduction in fossil fuel use that will be required to achieve this goal is more far-reaching than what has been adopted in the BAU scenario.*
Although I understand that this could not be included anymore in the paper, its needs a more extensive discussion. Please add a simple quantative analysis about what this could mean in terms of emission reductions of S and N from shipping as compared to the scenarios used.

General remark: For exceedance calculations, the CL's are used from the ICP M&M - CCE database that consists of nationally submitted CLs. Different countries however, submit CLs based on different receptors (surface waters, forest) and may use different CL methods (SMB or empirical) (noted in section 2.3). This leads to differences in CLs (and exceedances) that are not based on ecosystem sensitivity alone. This is for example clearly visible in Figure 5 where there is a change from AAE = 0 to AAE > 0 for CLAci over the Danish-German border which most likely due to differences in CL methods between Germany and Denmark. In figure 6, the AAE of CLeutN increases clearly going from Germany over the border to Denmark. Some discussion on this is needed to help the reader understand the figures better.

---

## Author Comment (AC1)

**Authors response for the review of "The impact of nitrogen and sulphur emissions from shipping on exceedances of critical loads in the Baltic Sea region."**

We would like to thank the two reviewers for all comments and suggestions that helped to improve the manuscript.

Our response to the Reviewers' comments is written below in italics:

**Review 1:**

**Abstract**

The abstract does not over the paper very well. Please add (at least) a sentence about the results of the sub-national analysis carried out for Sweden.

*-We have added the following sentences to the abstract:*

*"Geographically the impact of shipping emissions is unevenly distributed even within each country. This is illustrated by calculating CL exceedances for 21 Swedish counties. The impact on national level is driven by a few coastal counties where the impact of shipping is much higher than the national summary suggests."*

**Line 39. In the atmosphere ammonia reacts readily with both HNO3 and H2SO4 forming particulate ammonium sulphate and nitrate ((NH4)2SO4, NH4NO3).**

Change HNO3 and H2SO4 to H2SO4 and HNO3 to comply with the sequence used in the rest of the sentence and between the brackets.

*-Done*

**Line 49. but also eutrophication has decreased.**

This requires a reference

*-We have added the following reference: Engardt, M., Simpson, D., Schwikowski, M. and Granat, L.: Deposition of sulphur and nitrogen in Europe 1900–2050. Model calculations and comparison to historical observations. Tellus. Series B, Chemical and physical meteorology, 69 (1), 2017.*

**Line 59: with the ultimate aim reducing depositions below the CLs**

Please change to 'with the ultimate aim of reducing depositions below CLs'

*-Done*

**Line 79. limits both for S (SECA) and for NOx (NECA).**

Inconsistent. Why not S and N or SOx and NOx

*-Changed to SOx*

**Line 128. and include all merchant ships larger than 300 GT**

Please indicate what that means: how much of the total emissions is covered by limiting the data to ships of > 300GT?

*-The difference between CO2 emissions calculated from AIS signals of the IMO-registered ships (>300GT) and from all AIS signals is small, for year 2012 it was 2.5%. In recent years installations of AIS senders on small vessels, e.g. leisure boats are increasing and with them also this difference, in 2019 it was 15%.*

*We have added the following to the text: "These are based on position data from individual ships collected from AIS (Automatic Identification System) data and include all IMO registered merchant ships larger than 300GT. For smaller vessels however it is not mandatory to have AIS senders installed, and these are therefore not included. In recent years installations of AIS senders on small vessels have been increasing and Jalkanen (2020) found a difference of 15 % between the CO2 emissions calculated from AIS signals of the IMO-registered ships and the CO2 emissions calculated from all AIS signals."*

*Jalkanen, J.-P., 2020, Emissions from Baltic Sea shipping in 2006 - 2019, HELCOM Maritime 20/5-2.INF, 5-2 Emissions from Baltic Sea shipping in 2006 - 2019.pdf (helcom.fi).*

**Line 147-150. While for Germany the reason could be underestimation of NH3 emissions from agriculture, comparison of modelled and measured NH3 concentrations in Denmark and Poland shows overestimation by the model, indicating that the reason for underestimation of N deposition in these areas is rather availability of sulphuric and nitric acid or limited formation of particulate ammonium nitrate and sulphate.**

This reasoning is hard to follow. Why would one assume an underestimation of NH3 emissions in Germany being the cause for underestimating N depositions while as at the same time the NH3 concentrations in the adjacent Denmark are overestimated by the model which may, i.m.o. indicate overestimation of emissions of NH3. Please rephrase.

*-The text has been changed to:*

*"Comparison of modelled and measured NH3 gas concentrations in Denmark and Poland shows overestimation by the model, indicating that the reason for underestimation of N deposition in the southern part of the Baltic Sea region is the limited availability of sulfuric and nitric acid required for the formation of particulate ammonium nitrate and sulphates."*

*Note: measurements of NH3 concentrations had not been available from German stations during the period.*

**Line 169. Tier III for all ships built (keel laid) 2021 and later, operating in the region**

What does that addition mean, keel laid. Not clear. Is it needed, or is simply built 2021 and later enough?

*-Keel laid is the specified time in the regulation and is often used by SOLAS and IMO. Explanation added.*

**Line 171. In order to investigate impact of NECA**

Change to 'In order to investigate the impact of NECA'

*-Done*

**Line 234: Without introducing a NECA (scenario BAU-NoNECA) the contribution to N deposition would in median be more than twice as big as in the BAU case**

Unclear sentence. 'the contribution to N deposition would in median be more than twice as big'. Which median is meant here? The median of the depositions computed over all grid cells? Please clarify.

*-Yes, it is the median deposition on all grid cells in the modelled region, we have clarified this in the text. It now reads "the contribution to the N deposition on the grid cells within the modelled region would in median…"*

**Line 238: In the year 2012 deposition of S was still relatively high, reaching to >5 kg ha-1 yr-1 at the 1% of most impacted parts of the modelled area**

How is the most impacted part defined? As the area with the highest total deposition or the area with the highest deposition originating from shipping emissions or.... Please explain.

*-We meant it as the top 1% area with the highest total deposition of S in the modelled region, this has now been changed in the text.*

**Line 286: For the countries with the largest exceedances in 2012 there is a great improvement, and the impact of shipping in the year 2040 is rather insignificant**

'Great improvement ' and ' insignificant' So the improvement is from other emission reductions I assume? Please explain.

*-There is an improvement in the exceedances between 2012 and 2040 partly due to a decrease in shipping emissions of S, the S deposition attributed to shipping is very small in 2040.*

**Line 340. The highest average deposition of N (on acidification-sensitive ecosystems) acidification-sensitive ecosystems:**

how have these been defined? Do you assume all ecosystems for which CL's are submitted by Sweden are sensitive, or did you make a selection based on e.g. CLs ? Please explain.

*-The average is calculated on the area where Sweden has reported CL for acidity, there is no additional selection of low CL grid cells. Text clarified.*

**Line 428: The introduction of NECA will improve the situation in several of the Swedish counties, but more reductions might be necessary to further reduce the impact of shipping, there.**

This is a vague statement. More reduction might be needed. Make this more conclusive: what is needed in light of the AAE and how much can reducing shipping emissions contribute to that.

*-We have changed the text and are now more specific about the impact of shipping even when including NECA. The text now reads: "The introduction of NECA will improve the situation in several of the Swedish counties, but emissions from shipping will still contribute to the exceedances for several counties. In the five counties with the highest exceedances of CLeutN (both area and AAE) in the 2040 BAU scenario, shipping contributes to a mean of 18 % of the AAE."*

Tabell 12

**Line 452. The reduction in fossil fuel use that will be required to achieve this goal is more far-reaching than what has been adopted in the BAU scenario.**

Although I understand that this could not be included anymore in the paper, its needs a more extensive discussion.

Please add a simple quantative analysis about what this could mean in terms of emission reductions of S and N from shipping as compared to the scenarios used.

*-New potential abatement technologies to further reduce the fossil CO2 emissions have been screened e.g. by IMO 4th GHG study (Faber et al., 2020) and include a range of measures as e.g. use of fossil-based alternative fuels with carbon capture, use of fuels without fossil carbon, such as hydrogen, ammonia or synthetic and biomass carbon fuels with carbon capture, use of batteries and use of renewable energy, e.g. wind power and solar panels for auxiliary power. While some of these technologies are zero-emission regarding air pollutants (batteries, wind and solar power), others still have emissions of air pollutants, particularly of nitrogen (all combustion technologies, with or without carbon capture). Potential mixture of future technologies needed to meet the IMO 2050 target has not been presented yet and also emission factors for many of these not yet fully developed technologies are largely unknown. Quantitative estimate of potential impact of full implementation of the IMO 2050 target on scenarios presented in this study is therefore not currently possible.*

*Faber, J., Hanayama, S., Zhang, S., Pereda, P., Comer, B., Hauerhof, E., Schim van der Loeff, W., Smith, T., Zhang, Y., Kosaka, H., Adachi, M., Bonello, J., Galbraith, C., Gong, Z., Hirata, K., Hummels, D., Kleijn, A., Lee, D., Liu, Y., Lucchesi, A., Mao, X., Muraoka, E., Osipova, L., Qian, H., Rutherford, D., Suárez de la Fuente, S., Yuan, H., Velandia Perico, C., Wu, L., Sun, D., Yoo, D., and Xing, H.: The Fourth IMO GHG Study, London, UK., 2020.*

**General remark:** For exceedance calculations, the CL's are used from the ICP M&M - CCE database that consists of

nationally submitted CLs. Different countries however, submit CLs based on different receptors (surface waters, forest) and may use different CL methods (SMB or empirical)

(noted in section 2.3). This leads to differences in CLs (and exceedances) that are not based on ecosystem sensitivity alone. This is for example clearly visible in Figure 5 where there is a change from AAE = 0 to AAE > 0 for CLAci over the Danish-German border which most likely due to differences in CL methods between Germany and Denmark. In figure 6, the AAE of CLeutN increases clearly going from Germany over the border to Denmark. Some discussion on this is needed to help the reader understand the figures better.

*-Yes, countries have the liberty to compute CLs for ecosystems of their choice and also use chemical criteria that reflect their protection target. This (can) lead to jumps in CLs and their exceedances along country borders. We have mentioned this now in the paper in Section 2.3 and also in the discussion of Figure 6.*

**Review 2:**

GENERAL REMARKS The study investigates the impact of emissions from shipping on the acidification and eutrophication of soils and lakes in the Baltic Sea region. Exceedances of critical loads of ecosystems by the deposition of sulphur and nitrogen compounds from shipping are calculated for years 2012 and 2040, applying different emission control scenarios. The key finding of the study is that in year 2040 sulphur emissions from shipping have no further impact on the acidification of ecosystems if the rules of Sulphur Emission Control Areas (SECA) are applied. In contrast, nitrogen emissions from shipping still contribute to eutrophication, even if the rules of Nitrogen Emission Control Areas (NECA) are in force. The paper is very well written and structured, and the number and quality of figures is adequate. The number of tables is very large and the authors may consider moving some to supplementary material to increase readability. The study fits well into the scope of the journal and makes a significant contribution to the Special Issue on Shipping and the Environment. The manuscript will be suitable for publication once few issues as discussed in the following have been considered.

*-We have considered moving some of the tables into a supplementary, but we are inclined to keep them as is.*

**SPECIFIC COMMENT** The manuscript investigates the exceedances of critical loads of sulphur and nitrogen depositions. The authors use the expression "exceedance of critical loads" but refer to exceeded areas. I suggest using here the coherent wording "exceedance area". Then, the link between the two parameters is more evident. In the Introduction, the authors discuss the decrease in emissions of S and N species (line 44 – 49) and they present quantitative numbers for the achieved reduction. However, the reference year should be clearly defined. In the current version, the reference period is not clear.

*-We have calculated exceedances of CL both for area and AAE. In many places we mean both as there is no exceeded area without AAE. We have gone through the manuscript and clarified where the exceedance meant specifically exceeded area.*

*-The reference years for the numbers of emission decrease has been stated more clearly.*

MINOR ISSUES

**Line 34: Suggested re-phrasing: "takes predominantly place in the gas phase" instead of "takes … part in …".**

*-Changed*

**Line 45: Please clarify whether you mean "have decreased by 91% …" or "have decreased to 91% …".**

*-Done*

**Line 80: Do you refer to the fuel sulphur content by mass or by volume? Please specify.**

*-By mass, this has been clarified in the text.*

**Line 190: Parentheses are missing for the reference Posch et al. (2015).**

*-It's in the same parenthesis as the UNECE reference "(UNECE, 2004; see also Posch et al., 2015)".*

**Line 210: The Results Chapter is Chapter 3, not 2.**

*-Changed*

**Section 2.2 and Figure 3:** It is not readily obvious from the description of the shipping scenarios whether EEDI – Shipping is based on BAU with NECA implemented, or not. One clarifying sentence is appreciated, although this informant can be found in Table 2. Clarification would also help to better understand Fig. 4 and the other figures later in the manuscript which show results from the EEDI scenario.

*-We have changed the text at the end of Section 2.2 to make it clearer: "In order to investigate the impact of NECA and of the energy effectivization in the shipping sector two sensitivity scenarios were investigated: one which is the same as BAU but without the implementation of NECA (BAU-NoNECA) and another which is the same as BAU with the implementation of NECA but with the energy effectivization that just follows the IMO EEDI regulation (EEDI) (see Table 2)."*

**Table 2:** Is there a reason, why the first three columns contain values with one digit, while the last three columns contain two-digit values? Consistent presentation is suggested.

*-We have updated the table accordingly.*